# A CRISPR-Cas9 screen reveals a role for WD repeat-containing protein 81 (WDR81) in the entry of late penetrating viruses

Anthony J. Snyder📵, Andrew T. Abad📵, Pranav Danthi📵*

Department of Biology, Indiana University, Bloomington, Indiana, United States of America

* pdanthi@indiana.edu

## Abstract

Successful initiation of infection by many different viruses requires their uptake into the endosomal compartment. While some viruses exit this compartment early, others must reach the degradative, acidic environment of the late endosome. Mammalian orthoreovirus (reovirus) is one such late penetrating virus. To identify host factors that are important for reovirus infection, we performed a CRISPR-Cas9 knockout (KO) screen that targets over 20,000 genes in fibroblasts derived from the embryos of C57/BL6 mice. We identified seven genes (WDR81, WDR91, RAB7, CCZ1, CTSL, GNPTAB, and SLC35A1) that were required for the induction of cell death by reovirus. Notably, CRISPR-mediated KO of WD repeat-containing protein 81 (WDR81) rendered cells resistant to reovirus infection. Susceptibility to reovirus infection was restored by complementing KO cells with human WDR81. Although the absence of WDR81 did not affect viral attachment efficiency or uptake into the endosomal compartments for initial disassembly, it reduced viral gene expression and diminished infectious virus production. Consistent with the role of WDR81 in impacting the maturation of endosomes, WDR81-deficiency led to the accumulation of reovirus particles in dead-end compartments. Though WDR81 was dispensable for infection by VSV (vesicular stomatitis virus), which exits the endosomal system at an early stage, it was required for VSV-EBO GP (VSV that expresses the Ebolavirus glycoprotein), which must reach the late endosome to initiate infection. These results reveal a previously unappreciated role for WDR81 in promoting the replication of viruses that transit through late endosomes.

## Author summary

Viruses are obligate intracellular parasites that require the contributions of numerous host factors to complete the viral life cycle. Thus, the host-pathogen interaction can regulate cell death signaling and virus entry, replication, assembly, and egress. Functional genetic screens are useful tools to identify host factors that are important for establishing infection. Such information can also be used to understand cell biology. Notably, genome-scale CRISPR-Cas9 knockout screens are robust due to their specificity and the loss of host gene expression. Mammalian orthoreovirus (reovirus) is a tractable model

**Data Availability Statement:** All relevant data are within the manuscript and its Supporting Information files.

**Funding:** Research reported in this publication was supported by the National Institute of Allergy and Infectious Diseases of the National Institutes of Health under award numbers R01AI110637 and R03AI142013 (PD). The funders had no role in study design, data collection and analysis, decision to publish, or preparation of the manuscript.

**Competing interests:** The authors have declared that no competing interests exist.

system to investigate the pathogenesis of neurotropic and cardiotropic viruses. Using a CRISPR-Cas9 screen, we identified WD repeat-containing protein 81 (WDR81) as a host factor required for efficient reovirus infection of murine cells. Ablation of WDR81 blocked a late step in the viral entry pathway. Further, our work indicates that WDR81 is required for the entry of vesicular stomatitis virus that expresses the Ebolavirus glycoprotein.

## Introduction

As obligate intracellular parasites, viruses are dependent on host factors for many stages of virus replication. Classically, these factors were identified individually using biochemical methods and then biologically validated. More recently, functional genetic screens have been performed using genome-wide siRNA libraries, insertional mutagenesis of haploid cell lines, and CRISPR-Cas9 knockout screens [1–3]. Among these, CRISPR-Cas9 screening methods are likely the most robust because they are highly specific and typically result in the complete loss of gene expression due to genetic ablation. Curiously, however, screens for proviral host factors against the same virus in different cell lines can result in the identification of distinct host factors [4,5]. Such results indicate that while a subset of host factors is required to support replication of the virus in all cell types, the relative importance of other host factors for replication in different cell types varies. These results suggest that screens performed under different conditions are likely to reveal new host dependency factors.

Mammalian orthoreovirus (reovirus) is a segmented double-stranded RNA (dsRNA) virus with two concentric, protein shells [6]. Reovirus is used as a tractable model to investigate the pathogenesis of neurotropic and cardiotropic viruses. Multiple host and viral determinants that influence reovirus replication and host responses in human or murine cell lines also influence reovirus pathogenesis in a newborn mouse model [7–11]. As such, a number of proviral host factors have been identified for reovirus. These include cell surface molecules, such as cell surface glycans [12,13], Junctional adhesion molecule A (JAM-A) [14], Nogo Receptor 1 (NgR1) [15], and β1 integrin [16, 17], that directly engage reovirus particles. The steps following reovirus attachment are also dependent on host factors that deliver reovirus particles to late endosomes [18–22]. Within endosomes, low pH-dependent cathepsin B and L proteases mediate disassembly of the particle to generate an entry intermediate called infectious subvirion particle (ISVP) [23]. ISVPs undergo conformational transitions to form ISVP*s and deliver the genome-containing core particle (inner shell) into the host cytoplasm [24]. Cores transcribe viral mRNA that is translated and packaged into progeny cores generated from viral proteins. Following intraparticle dsRNA synthesis, the outer shell proteins are assembled onto the particle for subsequent release from infected cells. Surprisingly, with the exception of the TRiC chaperonin complex, which aids in reovirus outer shell assembly [25,26], few other host factors that act after ISVP formation have been identified.

In this work, we report the identification of host factors that are required for the infection of mouse embryo fibroblasts by reovirus using a CRISPR-Cas9 knockout screen. Among these, we find the BEACH (beige and Chediak–Higashi) and WD40 repeat-containing protein, WDR81, is required. WDR81 is a cytoplasmic protein that localizes with Early Endosome Antigen 1 (EEA1) and Lysosomal Associated Membrane Protein 1 (LAMP1) positive endolysosomal compartments [27,28]. WDR81 controls endosomal maturation by modulating phosphatidylinositol 3-phosphate (PtdIns3P) levels [28]. In the absence of WDR81, the transport of cargo destined for lysosomal enzyme mediated degradation is inhibited

[27,28]. We find that in WDR81-deficient cells, native reovirus particles are endocytosed and converted to ISVPs but fail to initiate further replication. Thus, WDR81 is required for the functional entry of ISVPs, which are generated in endosomes. Additionally, using vesicular stomatitis virus (VSV) and VSV that expresses the Ebolavirus glycoprotein (VSV-EBO GP), we find a role for WDR81 in the entry of other viruses that pass through the late endosome.

## Materials and methods

### Cells and viruses

Murine L929 (L) cells were grown at 37˚C in Joklik's Minimal Essential Medium (Lonza) supplemented with 5% fetal bovine serum (FBS) (Life Technologies), 2 mM L-glutamine (Invitrogen), 100 U/ml penicillin (Invitrogen), 100 µg/ml streptomycin (Invitrogen), and 25 ng/ml amphotericin B (Sigma-Aldrich). C57/BL6-derived mouse embryo fibroblasts (MEFs) (American Type Culture Collection) were grown at 37˚C in Dulbecco's Modified Eagle Medium (DMEM) (Gibco) supplemented with 10% FBS (Life Technologies) and 2 mM L-glutamine (Invitrogen). Baby Hamster Kidney-21 (BHK-21) cells (American Type Culture Collection) were grown at 37˚C in Minimal Essential Medium (Mediatech) supplemented with 10% FBS (Life Technologies), nonessential amino acids (Gibco), 2 mM L-glutamine (Invitrogen), 100 U/ml penicillin (Invitrogen), and 100 µg/ml streptomycin (Invitrogen). Human embryonic kidney (HEK) 293 FT cells (Thermo Fisher Scientific) were grown at 37˚C in DMEM (Gibco) supplemented with GlutaMAX (Gibco) and 10% FBS (Life Technologies). All reovirus experiments were performed with Type 3 Dearing from the Cashdollar laboratory (T3D$^{CD}$), which was provided by Dr. John Parker (Cornell University) [29], or with Type 1 Lang (T1L), which was generated by plasmid-based reverse genetics [30]. Vesicular stomatitis virus (VSV [GFP]) and Ebola Virus (EBO) glycoprotein (GP) expressing VSV (VSV-EBO GP [GFP]) were provided by Dr. Sean Whelan (Washington University School of Medicine in St. Louis). VSV (GFP) and VSV-EBO GP (GFP) were engineered to express green fluorescent protein (GFP) within infected cells [31].

### Reovirus propagation purification

Reovirus T3D$^{CD}$ and T1L were propagated and purified as previously described [32]. Briefly, L cells infected with second passage reovirus stocks were lysed by sonication. Virions were extracted from lysates using Vertrel-XF specialty fluid (Dupont). The extracted particles were layered onto 1.2- to 1.4-g/cm$^3$ CsCl step gradients. The gradients were then centrifuged at 187,000×$g$ for 4 h at 4˚C. Bands corresponding to purified virions (~1.36 g/cm$^3$) [33] were isolated and dialyzed into virus storage buffer (10 mM Tris-HCl, pH 7.4, 15 mM MgCl$_2$, and 150 mM NaCl). Following dialysis, the particle concentration was determined by measuring the optical density of the purified virion stocks at 260 nm (OD$_{260}$; 1 unit at OD$_{260}$ = 2.1×10$^{12}$ particles/ml) [33]. The purification of virions was confirmed by sodium dodecyl sulfate-polyacrylamide gel electrophoresis (SDS-PAGE) and Coomassie brilliant blue (Sigma-Aldrich) staining.

### Generation of reovirus infectious subviral particles (ISVPs)

Unlabeled and Alexa Fluor 488 (AF488)-labeled T3D$^{CD}$ or T1L virions (2×10$^{12}$ particles/ml) were digested with 200 µg/ml $N\alpha$-$p$-tosyl-L-lysine chloromethyl ketone (TLCK)-treated chymotrypsin (Worthington Biochemical) in a total volume of 100 µl for 20 min at 32˚C [34]. The reactions were then incubated on ice for 20 min and quenched by the addition of 1 mM phenylmethylsulfonyl fluoride (Sigma-Aldrich). The generation of ISVPs was confirmed by SDS-PAGE and Coomassie brilliant blue (Sigma-Aldrich) staining.

## VSV (GFP) and VSV-EBO GP (GFP) propagation

BHK-21 cells were adsorbed with stocks of VSV (GFP) or VSV-EBO GP (GFP) at 5 PFUs/cell for 1 h at room temperature. After 1 h, the cells were washed three times with PBS and incubated in growth medium for 24 h at 37˚C. Cell medium containing VSV (GFP) or VSV-EBO GP (GFP) was isolated and stored at -80˚C.

## Dynamic light scattering

Unlabeled and AF488-labeled T3D$^{CD}$ ($2\times10^{12}$ particles/ml) were analyzed using a Zetasizer Nano S dynamic light scattering system (Malvern Instruments). All measurements were made at room temperature in a quartz Suprasil cuvette with a 3.00-mm-path length (Hellma Analytics). For each sample, the size distribution profile was determined by averaging readings across 15 iterations.

## Reovirus plaque assays

Plaque assays to determine infectivity were performed [35]. Briefly, virions, ISVPs, or infected cell lysates were diluted into phosphate buffered saline (PBS) supplemented with 2 mM MgCl$_2$ (PBS$^{Mg}$). L cells grown in 6-well plates (Greiner Bio-One) were infected with 250 µl of the diluted virus for 1 h at room temperature. Following the viral attachment incubation, the monolayers were overlaid with 4 ml of serum-free medium 199 (Sigma-Aldrich) supplemented with 1% Bacto Agar (BD Biosciences), 10 µg/ml TLCK-treated chymotrypsin (Worthington, Biochemical), 2 mM L-glutamine (Invitrogen), 100 U/ml penicillin (Invitrogen), 100 µg/ml streptomycin (Invitrogen), and 25 ng/ml amphotericin B (Sigma-Aldrich). The infected cells were incubated at 37˚C, and plaques were counted at 5 days post infection.

## VSV (GFP) and VSV-EBO GP (GFP) plaque assays

Cell medium containing VSV (GFP) or VSV-EBO GP (GFP) was diluted into PBS$^{Mg}$. BHK-21 cells grown in 6-well plates (Greiner Bio-One) were infected with 250 µl of the diluted virus for 1 h at room temperature. Following the viral attachment incubation, the monolayers were overlaid with 2 ml of Minimal Essential Medium (Mediatech) supplemented with 5% FBS (Life Technologies), nonessential amino acids (Gibco), 2 mM L-glutamine (Invitrogen), 100 U/ml penicillin (Invitrogen), 100 µg/ml streptomycin (Invitrogen), and 1% low melt agarose (Fisher Scientific). The infected cells were incubated at 37˚C, and plaques were counted at 2 days post infection.

## Genome-scale CRISPR-Cas9 screen

The genome-scale CRISPR-Cas9 screen was performed using the mouse GeCKOv2 sgRNA library (Fig 1A) (Addgene) [36]. Briefly, lentivirus that harbored the mouse GeCKOv2 sgRNA library in the lentiCRISPRv2 vector (Addgene) was produced in HEK293FT cells (Thermo Fisher Scientific). MEF cells grown in 150-mm dishes (Greiner Bio-One) were transduced with the sgRNA library lentivirus at 0.3 infectious units/cell diluted in 8 µg/ml polybrene (EMD Millipore) [36], and the transduced cells were selected with 2 µg/ml puromycin (InvivoGen) in growth medium for 5 days at 37˚C. The cells were plated for full coverage of the library (pools A and B) [36]. The transduced and puromycin-resistant cells were then adsorbed with T3D$^{CD}$ at 5 plaque forming units (PFUs)/cell for 1 h at room temperature. After 1 h, the cells were washed three times with PBS and incubated in growth medium for 7 d at 37˚C. After 7 d, the virus-resistant cell population was transferred to fresh 150-mm dishes (Greiner Bio-One) and allowed to expand in growth medium supplemented with 2 µg/ml puromycin (InvivoGen)

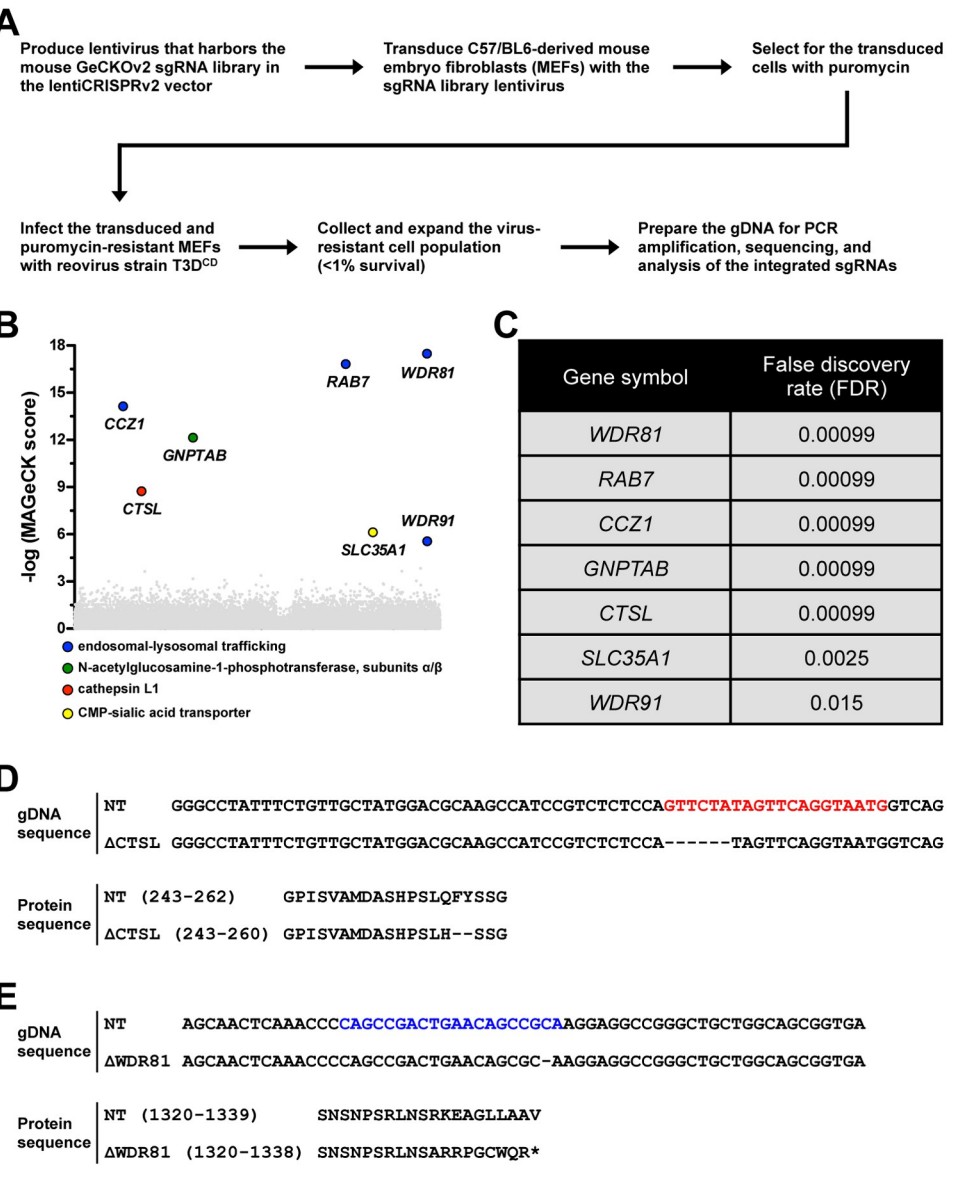

**Fig 1.** Genome-scale CRISPR-Cas9 knockout screen. (A) Strategy and workflow to identify host factors that are important for reovirus infection. (B and C) Screen results for gene-targeting sgRNAs that were enriched in the virus-resistant cell population. WDR81 sgRNA sequences were highly represented in the recovered clones (n = 2 biological replicates). (D and E) Genomic DNA and protein sequences for CTSL- (D) and WDR81- (E) deficient cell lines. The CTSL- and WDR81-targeting sgRNA sequences are bolded in red and blue, respectively.

for 1–2 d at 37°C. Following expansion, the genomic DNA (gDNA) was prepared using the *Quick*-DNA Midiprep Plus Kit (Zymo Research), and the integrated sgRNAs were amplified by polymerase chain reaction (PCR) [36]. The PCR products were isolated by agarose gel electrophoresis and subsequent gel extraction using the QIAquick Gel Extraction Kit (Qiagen). The purified PCR products were then processed and sequenced using the NextSeq 75 –High Output (82 cycles in read 1, 8 cycles in index 1, and 8 cycles in index 2 SE reads) (Illumina). The sequencing data was analyzed using the Model-Based Analysis of Genome-wide CRISPR/Cas9 Knockout (MAGeCK) system [37].

## Generation of CRISPR modified cell lines

Non-targeting (NT), Cathepsin L (CTSL), and WD repeat-containing protein 81 (WDR81) variant cell lines were generated using CRISPR technology. Briefly, the sgRNA sequences representing NT, CTSL, and WDR81 targets were cloned into the lentiCRISPRv2 vector (Addgene) using Golden Gate assembly reactions [36]. The primers used for cloning were as follows: NT ([5'-CACCGCTGAAAAAGGAAGGAGTTGA-3'] and [5'-AAACTCAACTCCT TCCTTTTTCAGC-3']), CTSL ([5'-CACCGCATTACCTGAACTATAGAAC-3'] and [5'-AAA CGTTCTATAGTTCAGGTAATGC-3']), and WDR81([5'-CACCGCAGCCGACTGAACAG CCGCA-3'] and [5'-AAACTGCGGCTGTTCAGTCGGCTGC-3']). Lentivirus that harbored the NT, CTSL, or WDR81 sgRNA in the lentiCRISPRv2 vector (Addgene) was produced in HEK293FT cells (Thermo Fisher Scientific). MEF cells grown in 6-well plates (Greiner Bio-One) were transduced with the lentivirus at 0.3 infectious units/cell diluted in 8 μg/ml polybrene (EMD Millipore) as previously described, and the transduced cells were selected with 2 μg/ml puromycin (InvivoGen) in growth medium for 5 days at 37˚C. The transduced and puromycin-resistant cells were then transferred to 150-mm dishes (Greiner Bio-One) by limiting-dilution and allowed to expand in growth medium supplemented with 2 μg/ml puromycin (InvivoGen) for 3–5 d at 37˚C. Single colonies were isolated using glass cloning cylinders (Sigma-Aldrich), transferred to 6-well plates (Greiner Bio-One), and allowed to expand in growth medium supplemented with 2 μg/ml puromycin (InvivoGen) for 3–5 d at 37˚C. To identify and validate isolates that contain a modified CTSL or WDR81, gDNA was prepared using the *Quick*-DNA Midiprep Plus Kit (Zymo Research), and the gene encoding regions were amplified by PCR. The primers used for amplification were as follows: CTSL ([5'-GGACA TAGTGTCCATAAGTCCTC-3'] and [5'-CTTCCGTGGGATTCTCATTTCCTC-3']) and WDR81 ([5'-GTAGGAAGACCACAGATTCAGTG-3'] and [5'-CCAGCTGAGTGATAAGAGCAGTAC-3']). The PCR products were isolated by agarose gel electrophoresis and subsequent gel extraction using the QIAquick Gel Extraction Kit (Qiagen). The purified PCR products then were processed and sequenced. The primers used for sequencing were as follows: CTSL ([5'-GGTTCC AGAACCCATATTAGACAG-3'] and [5'-GAGTTCGCTGTGGCTAATGACAC-3']) and WDR81 ([5'-CAAGACCTCGTGGCTAATGC-3'] and [5'-CCACATTGCCTACCTGTATGGAG-3']).

## Generation of WDR81 complementing cell lines

NT-empty, ΔWDR81-empty, and ΔWDR81-WDR81 cells were generated using a lentivirus transduction and selection strategy. Briefly, the lentiviral vectors pHRSIN.pSFFV MCS(+) pSV40 Blast and pHRSIN.pSFFV 2xHA-WDR81 pSV40 Blast, which encode empty and human WDR81 fused to two hemagglutinin (HA) tags, respectively, were provided by Dr. Paul Lehner (University of Cambridge) [27]. These vectors were used to produce lentivirus in HEK293FT cells (Thermo Fisher Scientific). NT and ΔWDR81 cells grown in 6-well plates (Greiner Bio-One) were transduced with the lentiviruses at 0.3 infectious units/cell diluted in 8 μg/ml polybrene (EMD Millipore) as previously described, and the transduced cells were selected with 2 μg/ml puromycin (InvivoGen) and 8 μg/ml blasticidin (InvivoGen) in growth medium for 5 days at 37˚C. The transduced and puromycin- and blasticidin-resistant cells were then transferred to 150-mm dishes (Greiner Bio-One) by limiting-dilution and allowed to expand in growth medium supplemented with 2 μg/ml puromycin (InvivoGen) and 8 μg/ ml blasticidin (InvivoGen) for 3–5 d at 37˚C. Single colonies were isolated using glass cloning cylinders (Sigma-Aldrich), transferred to 6-well plates (Greiner Bio-One), and allowed to expand in growth medium supplemented with 2 μg/ml puromycin (InvivoGen) and 8 μg/ml blasticidin (InvivoGen) for 3–5 d at 37˚C. To identify and validate isolates that express 2xHA-WDR81, NT-empty, ΔWDR81-empty, and ΔWDR81-WDR81 cells were grown in

96-well plates (Greiner Bio-One). At ~90% confluency, the cells were fixed with 100 μl of methanol for 30 min at -20˚C, washed three times with PBS, and blocked with PBS supplemented with 2.5% BSA and 0.25% Triton X-100 (TX-100) for 30 min at room temperature. The blocked cells were incubated with an α-HA-biotin primary antibody (Sigma-Aldrich) diluted 1:1,000 into PBS supplemented with 0.25% TX-100 for 30 min at 37˚C. The cells were washed three times with PBS followed by incubation with an IRDye 800CW Streptavidin secondary antibody (LI-COR) diluted 1:5,000 into PBS supplemented with 0.25% TX-100 for 30 min at 37˚C. The cells were then washed three times with PBS and total cells were labeled with a 1:1,000 dilution of DRAQ5 (Cell Signaling Technology) for 5 min at 4˚C. The cells were washed three times with PBS and scanned using an Odyssey imaging system (LI-COR).

## Sequence analysis

The reference sequences for *Mus musculus* (house mouse) derived CTSL and WDR81 and for *Homo sapiens* (human) derived WDR81 were obtained from the National Center for Biotechnology Information database.

## Assessment of reovirus-induced cell death

NT, ΔCTSL, ΔWDR81, NT-empty, ΔWDR81-empty, and ΔWDR81-WDR81 cells were grown in 96-well plates (Greiner Bio-One). The cells were adsorbed with T3D$^{CD}$ at 5 PFUs/cell for 1 h at room temperature. After 1 h, the cells were washed three times with PBS and incubated in growth medium for 24 or 48 h at 37˚C. Cell viability was measured using the CellTiter-Glo Luminescent Cell Viability Assay (Promega) by following the manufacturer's instructions. Relative cell viability was calculated by dividing the luminescent values of infected cells by the luminescent values of uninfected cells.

## Assessment of VSV (GFP)- and VSV-EBO GP (GFP)-induced cell death

NT, ΔCTSL, and ΔWDR81 cells were grown in 96-well plates (Greiner Bio-One). The cells were adsorbed with VSV (GFP) or VSV-EBO GP (GFP) at 5 PFUs/cell for 1 h at room temperature. After 1 h, the cells were washed three times with PBS and incubated in growth medium for 18 or 26 h at 37˚C. Cell viability was measured using the CellTiter-Glo Luminescent Cell Viability Assay (Promega) following the manufacturer's instructions. Relative cell viability was calculated by dividing the luminescent values of infected cells by the luminescent values of uninfected cells.

## Assessment of reovirus cell attachment by indirect immunofluorescence

Quantification of reovirus attachment was performed as previously described [38]. Briefly, NT, ΔCTSL, and ΔWDR81 cells grown in 96-well plates (Greiner Bio-One) were chilled for 15 min at 4˚C. The chilled cells were adsorbed with T3D$^{CD}$ at $1.0\times10^{6}$ virions/cell or $1.0\times10^{6}$ ISVPs/cell for 1 h at 4˚C. After 1 h, the cells were washed three times with chilled PBS and blocked with PBS supplemented with 5% bovine serum albumin (PBS-BSA) for 10 min at 4˚C. The cells were then incubated with an α-reovirus primary antibody [39] diluted 1:2,500 into PBS-BSA for 30 min at 4˚C. The cells were washed three times with PBS-BSA followed by incubation with an IRDye 800CW secondary antibody (LI-COR) diluted 1:1,000 into PBS-BSA for 30 min at 4˚C. After two washes with PBS-BSA, total cells were labeled with a 1:1,000 dilution of DRAQ5 (Cell Signaling Technology) for 5 min at 4˚C. The cells were washed three times with PBS-BSA and then fixed with 4% formaldehyde for 20 min at room temperature. The fixed plates were scanned using an Odyssey imaging system (LI-COR). The binding index was quantified by the ratio of green (attached virus) and red (total cells) fluorescence using Image Studio Lite software (LI-COR).

## Assessment of reovirus in-cell particle disassembly

NT, ΔCTSL, and ΔWDR81 cells grown in 24-well plates (Greiner Bio-One) were chilled for 15 min at 4˚C. The chilled cells were adsorbed with T3D$^{CD}$ at $1.0 \times 10^3$ virions/cell for 1 h at 4˚C. After 1 h, the cells were washed three times with PBS and incubated in growth medium at 37˚C. At the indicated times post infection, the infected monolayers were washed three times with PBS and lysed with RIPA buffer (50 mM Tris-HCl, pH 7.5, 50 mM NaCl, 1% Triton X-100, 1% DOC, 0.1% SDS, and 1 mM EDTA). The cell lysates were solubilized in reducing SDS sample buffer and analyzed by SDS-PAGE. The levels of reovirus μ1C/δ and the PSTAIR epitope of the host protein Cdk1 were determined by Western blot using an anti-reovirus primary antibody and an anti-PSTAIR primary antibody (Sigma-Aldrich), respectively.

## Assessment of reovirus infectivity by RNA transcription

NT, ΔCTSL, and ΔWDR81 cells were grown in 6-well plates (Greiner Bio-One). The cells were adsorbed with T3D$^{CD}$ at 5 PFUs/cell for 1 h at room temperature. After 1 h, the cells were washed three times with PBS and incubated in growth medium for 6 or 18 h at 37˚C. The cells were then washed three times with PBS and lysed with Tri Reagent (Molecular Research Center). Total RNA was extracted from the cell lysates by following the manufacturer's instructions. For quantitative reverse transcription-PCR (qRT-PCR), 1 μg of RNA was reverse transcribed using the High-Capacity cDNA Reverse Transcription kit (Applied Biosystems) following the manufacturer's instructions and gene specific primers against the T3D$^{CD}$ S1 gene segment ([5'-TGGCGAGATTATTCCCTGAC-3']) and the murine glyceraldehyde-3-phosphate dehydrogenase (GAPDH) mRNA ([5'-GGATGCAGGGATGATGTTCT-3']). The cDNA was diluted 1:10 into ultrapure H$_2$O, mixed with forward and reverse detection primers (T3D$^{CD}$ S1 forward [5'-TACGCGTTGATCACGACAAT-3'] and T3D$^{CD}$ S1 reverse [5'-TGGCGAGATTATTCCCTGAC-3'] or GAPDH forward [5'-ACCCAGAAGACTGTGGATGG-3'] and GAPDH reverse [5'-GGATGCAGGGATGATGTTCT-3']) and SYBR Select Master Mix (Applied Biosystems), and then subjected to PCR using the StepOnePlus Real-Time PCR system (Applied Biosystems). Multiple qRT-PCR measurements were made for each sample. $\Delta C_T$ values were calculated by subtracting the threshold cycle ($C_T$) values of GAPDH from the $C_T$ values of the T3D$^{CD}$ S1 gene segment. Levels of RNA transcription with respect to the control sample was quantified using the $\Delta\Delta C_T$ method [40].

## Assessment of reovirus infectivity by initiation of protein synthesis

NT, ΔCTSL, and ΔWDR81 cells were grown in 24-well plates (Greiner Bio-One). The cells were adsorbed with T3D$^{CD}$ at 5 PFUs/cell for 1 h at room temperature. After 1 h, the cells were washed three times with PBS and incubated in growth medium at 37˚C. At the indicated times post infection, the infected monolayers were washed three times with PBS and lysed with RIPA buffer (50 mM Tris-HCl, pH 7.5, 50 mM NaCl, 1% Triton X-100, 1% DOC, 0.1% SDS, and 1 mM EDTA). The cell lysates were solubilized in reducing SDS sample buffer and analyzed by SDS-PAGE. The levels of reovirus σNS and the PSTAIR epitope of the host protein CDK1 were determined by Western blot using an anti-σNS primary antibody [41] and an anti-PSTAIR primary antibody (Sigma-Aldrich), respectively.

## Assessment of VSV (GFP) and VSV-EBO GP (GFP) infectivity by initiation of protein synthesis

NT, ΔCTSL, and ΔWDR81 cells were grown in 6-well plates (Greiner Bio-One). The cells were adsorbed with VSV (GFP) or VSV-EBO GP (GFP) at 5 PFUs/cell for 1 h at room temperature.

After 1 h, the cells were washed three times with PBS and incubated in growth medium at 37˚C in an IncuCyte S3 Live-Cell Analysis System (Sartorius). At the indicated times post infection, three images per well were acquired using a X10 objective and green (excitation [440–480 nm], emission [504–544 nm]) and phase channels. The images were analyzed using the IncuCyte S3 Live-Cell Analysis System (Sartorius) software and plotted as GFP positive cells (per image) / Percent cell confluency versus Time post infection (h).

## Assessment of reovirus infectivity by indirect immunofluorescence

NT, ΔCTSL, ΔWDR81, NT-empty, ΔWDR81-empty, and ΔWDR81-WDR81 cells were grown in 96-well plates (Greiner Bio-One). The cells were adsorbed with T3D$^{CD}$ at 5 PFUs/cell for 1 h at room temperature. After 1 h, the cells were washed three times with PBS and incubated in growth medium for 18 h at 37˚C. The infected monolayers were fixed with 100 μl of methanol for 30 min at -20˚C, washed three times with PBS, and blocked with PBS supplemented with 2.5% BSA and 0.25% Triton X-100 (TX-100) for 30 min at room temperature. The blocked cells were incubated with an α-reovirus primary antibody diluted 1:5,000 into PBS supplemented with 0.25% TX-100 for 30 min at 37˚C. The cells were washed three times with PBS followed by incubation with an IRDye 800CW secondary antibody (LI-COR) diluted 1:5,000 into PBS supplemented with 0.25% TX-100 for 30 min at 37˚C. The cells were then washed three times with PBS and total cells were labeled with a 1:1,000 dilution of DRAQ5 (Cell Signaling Technology) for 5 min at 4˚C. The cells were washed three times with PBS and scanned using an Odyssey imaging system (LI-COR). Infectivity index was quantified by the ratio of green (virus infected cells) and red (total cells) fluorescence using Image Studio Lite software (LI-COR).

## Single step reovirus growth assay

NT, ΔCTSL, ΔWDR81, NT-empty, ΔWDR81-empty, and ΔWDR81-WDR81 cells were grown in 6-well plates (Greiner Bio-One). The cells were adsorbed with T3D$^{CD}$ or T1L at 5 PFUs/cell for 1 h at room temperature. After 1 h, the cells were washed three times with PBS and incubated in growth medium for 18 or 24 h at 37˚C. The infected cells were lysed by two freeze-thaw cycles and the amount of infectious virus produced was measured using plaque assay.

## Single-step VSV (GFP) and VSV-EBO GP (GFP) growth assay

NT, ΔCTSL, and ΔWDR81 cells were grown in 6-well plates (Greiner Bio-One). The cells were adsorbed with VSV (GFP) or VSV-EBO GP (GFP) at 5 PFUs/cell for 1 h at room temperature. After 1 h, the cells were washed three times with PBS and incubated in growth medium for 10 h (VSV [GFP]) or 18 h (VSV-EBO GP [GFP]) at 37˚C. Cell media containing VSV (GFP) or VSV-EBO GP (GFP) were collected, and the virus titer was determined by plaque assay.

## Conjugation of Alexa Fluor 488 to purified reovirus

Purified reovirus was labeled using the Alexa Fluor 488 Protein Labeling Kit (Invitrogen). Briefly, T3D$^{CD}$ virions ($9 \times 10^{13}$ particles/ml) were diluted into fresh 50 mM sodium bicarbonate (pH 8.5) and incubated in one vial of AF488 carboxylic acid, tetrafluorophenyl ester for 90 min at room temperature in the dark. After 90 min, the reaction was quenched by the addition of virus storage buffer (10 mM Tris-HCl, pH 7.4, 15 mM MgCl$_2$, and 150 mM NaCl), and the labeled virions were layered onto 1.2- to 1.4-g/cm$^3$ CsCl step gradients. The gradients were then centrifuged at 187,000×$g$ for 4 h at 4˚C. Bands corresponding to purified and labeled virions (~1.36 g/cm$^3$) [33] were isolated and dialyzed into virus storage buffer (10 mM Tris-HCl,

pH 7.4, 15 mM $MgCl_2$, and 150 mM NaCl). Following dialysis, the particle concentration was determined by measuring the optical density of the purified virion stocks at 260 nm ($OD_{260}$; 1 unit at $OD_{260}$ = $2.1 \times 10^{12}$ particles/ml) [33]. The purification of labeled virions was confirmed by SDS-PAGE and scanning for AF488 fluorescence using a ChemiDoc MP Imaging System (Bio-Rad).

## Assessment of reovirus localization within an infected cell

NT, ΔCTSL, and ΔWDR81 cells were grown on a four chambered polystyrene vessel tissue culture treated glass slide (Corning). The cells were adsorbed with $T3D^{CD}$ (AF488) at 10,000 particles/cell for 1 h at 4°C. After 1 h, the chilled cells were washed three times with PBS and incubated in growth medium for 2 h at 37°C. The infected monolayers were then fixed with 200 μl of 3.7% formaldehyde for 10 min at room temperature and washed three times with PBS. When indicated, the fixed cells were stained with 200 μl of an anti-EEA1 (Cell Signaling), anti-Rab7 (Cell Signaling), or anti-LAMP1 (Abcam) primary antibody and with 200 μl of 1.0 μg/ml 4',6-diamidino-2-phenylindole (DAPI; Invitrogen) diluted in PBS. The fixed and stained cells were washed with PBS and mounted using Aqua-Poly/Mount (Polysciences). Confocal images of infected cells were acquired using a Leica SP8 Scanning Confocal Microscope controlled by Leica-X software. The images were obtained using a X63 oil-immersion objective and White Light and 405 nm lasers and HyD detectors. Three-dimensional image stacks were acquired by recording sequential sections through the z-axis. All images were processed with ImageJ software (two-dimensional maximum intensity projections) [42]. Colocalization between AF488 and EEA1, Rab7, or LAMP1 was quantified using Just Another Colocalization Plugin (JACoP) [43].

## Statistical analyses

Unless otherwise noted, the results from all experiments represent three or four biological replicates. Horizontal bars indicate the means. Error bars indicate the standard deviations. *P* values were calculated using one-way analysis of variance with Bonferroni Correction or Student's *t* test (two-tailed, unequal variance assumed). For comparison of titers, two criteria were used to assign significance: *P* value of $\leq 0.05$ and difference in titer of $\geq 1$ log(PFU/ml) unit.

# Results

## CRISPR-Cas9 screen uncovers host factors that are required for reovirus infection

Toward the goal of identifying as yet unknown host factors that are required for reovirus infection of host cells, we undertook a genome-scale CRISPR-Cas9 positive selection screen in murine embryo fibroblasts (MEFs) derived from C57/BL6 mice. We selected MEFs for this screen because unlike transformed cells that are typically used to propagate and investigate reovirus infection, MEFs have a genomic makeup that resembles the normal diploid genome of the murine host. Additionally, multiple host factors that are important for reovirus infection and the host response have been identified or studied using MEFs [8,9,23,44,45]. To complete this screen, MEFs were transduced with lentiviruses derived from the Mouse CRISPR Knockout Pooled Library (GeCKO v2) in a one vector system, which encodes both Cas9 and a single-guide RNA (sgRNA). Transduced cells, which were selected with puromycin, were challenged with reovirus strain T3D-Cashdollar ($T3D^{CD}$). $T3D^{CD}$ is significantly more cytotoxic than other laboratory isolates of T3D [29]. A small percentage of reovirus-resistant cells were

obtained and subsequently expanded. The representation of sgRNAs in the transduced pool (prior to reovirus infection) and the reovirus-resistant pool (obtained following reovirus infection) was quantified by deep sequencing PCR products that were amplified from the genomic DNA (gDNA) (Fig 1A).

The GeCKO v2 library contains ~100,000 unique sgRNAs that target the mouse genome. Each gene is represented by 6 unique sgRNAs. The sequencing results were subjected to MAGeCK analysis, which ranks positively selected hits based on the number of sgRNAs enriched for each gene and the extent of enrichment of these sgRNAs following selection [37]. Using a false discovery rate (FDR) cut-off of ≤ 0.05, MAGeCK identified 7 significant hits (Fig 1B and 1C). These hits include SLC35A1 and CTSL, whose gene products are required for reovirus infection. SLC35A1 is necessary for the cell surface expression of sialic acid, which serves as a receptor for type 3 reovirus strains, such as T3D [46]. CTSL encodes the endosomal protease, cathepsin L, which is required for capsid disassembly, a necessary step in reovirus cell entry [23]. The five additional hits that were identified in our screen have not been directly tested through gene knockdown or knockout experiments to evaluate their function in reovirus infection. For our current study, we characterized the function of a poorly studied BEACH (beige and Chediak–Higashi)- and WD40 repeat-containing protein, WDR81. Interestingly, WDR81 interacts with WDR91 [27,28], which was also identified in our work (Fig 1B and 1C) and in another screen for proviral factors of reovirus infection [46].

To evaluate the function of WDR81 in reovirus infection, we used CRISPR-Cas9 editing to generate WDR81-deficient MEFs. Sequencing the gDNA of ΔWDR81 MEFs indicated that gene editing resulted in the introduction of a premature stop codon in the WDR81 open reading frame (Fig 1E). ΔCTSL MEFs, which were similarly generated by CRISPR-Cas9 editing, contained an in-frame deletion in the CTSL gene. ΔCTSL MEFs were used as a positive control cell line that is unable to support reovirus infection (Fig 1D). We first measured viability of control non-targeting (NT), ΔCTSL, and ΔWDR81 cells following infection with T3D. We found that the viability of NT cells decreased at 48 h post infection (hpi). In contrast, ΔCTSL, and ΔWDR81 remained resistant to reovirus-induced cell death (Fig 2A). These results validate our CRISPR-Cas9 screen. Further, our data suggest a previously unidentified role for WDR81 in impacting reovirus-induced cell death or a role for WDR81 in supporting events in reovirus infection, which ultimately affect reovirus-induced cell death.

## Reovirus fails to launch infection in the absence of WDR81

Reovirus initiates infection through the engagement of one or more cell surface receptors (Fig 2B). To determine whether the absence of WDR81 influences reovirus attachment, NT, ΔCTSL, and ΔWDR81 cells were adsorbed with reovirus, and the attached reovirus was quantified by indirect immunofluorescence. Our results demonstrate equivalent attachment of reovirus to each cell type (Fig 2C). These data indicate that ΔWDR81 does not influence reovirus attachment. Attached and internalized reovirus particles traffic through the endosomal-lysosomal system where they are acted on by luminal proteases, such as cathepsin L and B [23]. These proteases degrade the viral σ3 outer-capsid protein and cleave the μ1 protein to produce the particle-associated δ and φ fragments to form ISVPs (Fig 2B). As a marker for ISVP formation, we monitored the cleavage of μ1 into δ in NT, ΔCTSL, and ΔWDR81 cells. We found that δ was rapidly formed from μ1 in NT and ΔWDR81 cells (Fig 2D). In contrast, μ1 remained uncleaved in ΔCTSL cells due to the absence of functional cathepsin L. These data indicate that ΔWDR81 is not required for ISVP formation. Further, these results demonstrate that the absence of Cathepsin L and WDR81 renders cells resistant to reovirus-induced cell death via different mechanisms.

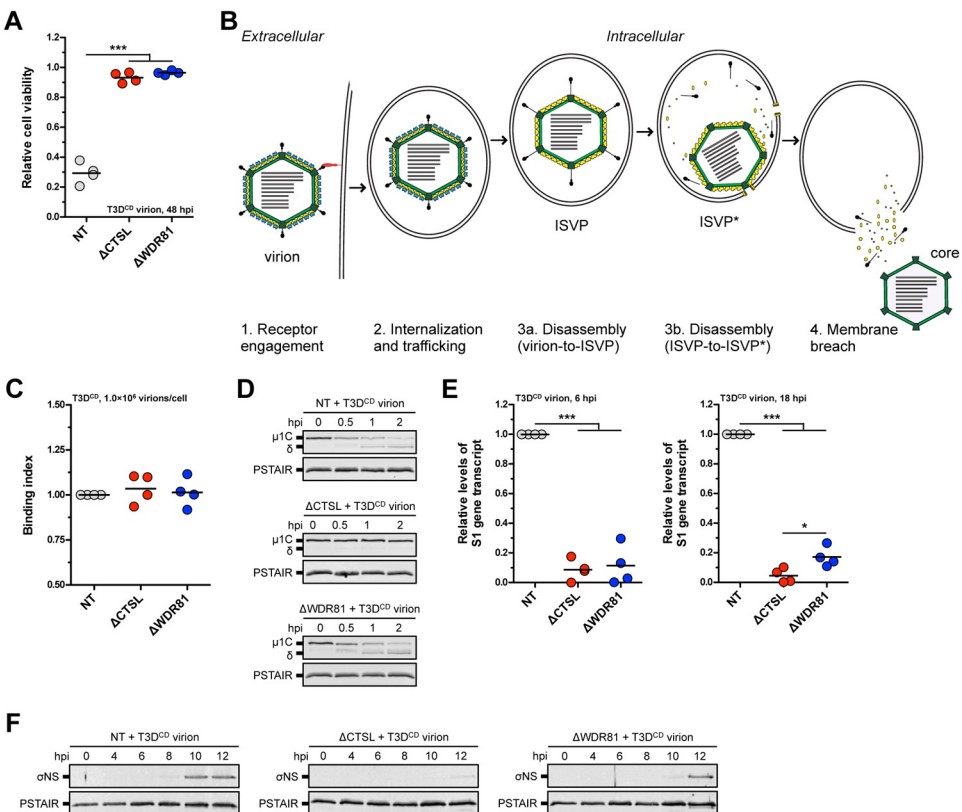

**Fig 2. Infection of WDR81-deficient cells by T3D<sup>CD</sup>.** (A) Induction of cell death. NT, ΔCTSL, and ΔWDR81 cells were infected with T3D$^{CD}$ virions at 5 PFUs/cell. At 48 h post infection (hpi), the relative cell viability was measured using the CellTiter-Glo Luminescent Cell Viability Assay. Horizontal bars indicate the means; ***, $P \leq 0.0005$ (n = 4 biological replicates). (B) Schematic of the reovirus entry pathway. (C) Viral attachment. NT, ΔCTSL, and ΔWDR81 cells were adsorbed with T3D$^{CD}$ virions at $1.0 \times 10^6$ particles/cell, and the binding index was measured using indirect immunofluorescence. Horizontal bars indicate the means (n = 4 biological replicates). (D) Viral disassembly (virion-to-ISVP). NT, ΔCTSL, and ΔWDR81 cells were infected with T3D$^{CD}$ virions at $1.0 \times 10^3$ particles/cell. At the timepoints indicated in the figure, the levels of the reovirus μ1C and δ proteins and the PSTAIR epitope of the host protein Cdk1 were measured by Western blot. The migration of μ1C, δ, and PSTAIR are indicated on the left (n = 3 biological replicates; results from 1 representative experiment are shown). (E) Viral RNA transcription. NT, ΔCTSL, and ΔWDR81 cells were infected with T3D$^{CD}$ virions at 5 PFUs/cell. At 6 and 18 hpi, the levels of the reovirus S1 gene transcript were measured using quantitative reverse transcription-PCR. Horizontal bars indicate the means; *, $P \leq 0.05$, ***, $P \leq 0.0005$ (n = 4 biological replicates). (F) Initiation of viral protein synthesis. NT, ΔCTSL, and ΔWDR81 cells were infected with T3D$^{CD}$ virions at 5 PFUs/cell. At the timepoints indicated in the figure, the levels of the reovirus σNS protein and the PSTAIR epitope of the host protein Cdk1 were measured by Western blot. The migration of σNS and PSTAIR are indicated on the left (n = 3 biological replicates; results from 1 representative experiment are shown).

Successful disassembly of reovirus results in the delivery of the reovirus core particle to the host cytoplasm (Fig 2B) [6]. Cytoplasmically localized cores become transcriptionally active and synthesize viral mRNA using a viral RNA-dependent RNA polymerase, its cofactors, and capping machinery that are all contained within the core. The accumulation of viral S1 mRNA at 6 and 18 hpi was measured using RT-qPCR. We found that compared to NT cells, the levels of the S1 gene transcript in ΔCTSL and ΔWDR81 were substantially lower at each time point (Fig 2E). These data suggest that reovirus fails to efficiently launch infection in the absence of WDR81. Consistent with lower mRNA expression, we found that reovirus protein synthesis, as measured by the expression of the non-structural σNS protein, was not detected in ΔCTSL cells and is delayed in ΔWDR81 cells (Fig 2F). Reovirus infectivity was also measured by

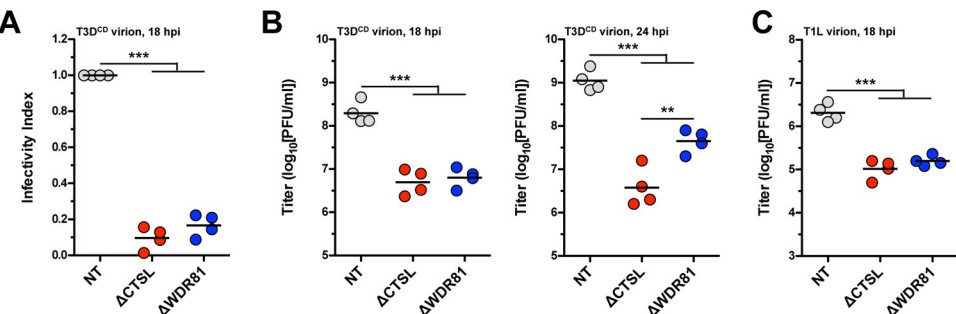

**Fig 3. Infectivity of T3D$^{CD}$ and T1L virions in WDR81-deficient cells.** (A) Levels of infectivity. NT, ΔCTSL, and ΔWDR81 cells were infected with T3D$^{CD}$ virions at 5 PFUs/cell. At 18 hpi, the infectivity index was measured using indirect immunofluorescence. Horizontal bars indicate the means; ***, $P \leq 0.0005$ (n = 4 biological replicates). (B and C) Production of infectious virus. NT, ΔCTSL, and ΔWDR81 cells were infected with T3D$^{CD}$ (B) or T1L (C) virions at 5 PFUs/cell. At 18 and 24 hpi, the amount of infectious virus produced was measured using plaque assay. Horizontal bars indicate the means; **, $P \leq 0.005$, ***, $P \leq 0.0005$ (n = 4 biological replicates).

indirect immunofluorescence. Similar to the results shown above, reovirus infectivity was significantly lower in ΔCTSL and ΔWDR81 cells (Fig 3A). Importantly, the reductions in RNA synthesis and protein synthesis were sufficient to produce a diminishment in virus production (Fig 3B). In comparison to NT cells, virus production was strongly blocked in ΔCTSL cells. In contrast, an intermediate, but still significantly lower level of virus, was produced from ΔWDR81 cells. Since virus production from ΔWDR81 cells remained low over an extended time frame, these data suggest that infection is not simply slower in the absence of WDR81.

The output of reovirus strain Type 1 Lang (T1L) was similarly reduced from ΔWDR81 cells (Fig 3C). Together, our data indicate that WDR81 is required for efficient reovirus replication. These results also suggest that a step after ISVP formation requires WDR81 (Fig 2D). Because viral RNA synthesis is the first step that is affected in the absence of WDR81, either RNA synthesis itself or events between ISVP formation and RNA synthesis, appear to require WDR81 (Fig 2E). Such steps could include ISVP-to-ISVP* conversion, pore formation, or core delivery into the host cytoplasm [47]. Because WDR81 directly affects the early events in infection, the increased survival of ΔWDR81 cells to reovirus infection is at least in part related to the inability of these cells to support reovirus infection. For the purpose of this study, we did not evaluate if the absence of WDR81 influences cell death independent of the capacity to allow reovirus infection.

## Complementation of ΔWDR81 cells with human WDR81 restores sensitivity to reovirus infection

To ensure that the observed defect in reovirus infection was due to the genetic ablation of WDR81, we transduced ΔWDR81 cells with an empty lentivirus or a lentivirus that confers expression of 2xHA-tagged human WDR81 (hu-WDR81), which shares 88% homology with murine WDR81. Expression of hu-WDR81 in ΔWDR81-WDR81 cells, but not in ΔWDR81-empty cells, was confirmed by indirect immunofluorescence (Fig 4A). The complemented cells were challenged with reovirus to determine whether the expression of hu-WDR81 restores sensitivity to reovirus-induced cell death. As expected, NT-empty cells succumbed to infection in a time-dependent manner, whereas ΔWDR81-empty cells remained resistant. In contrast, ΔWDR81-WDR81 succumbed to reovirus infection, albeit to a slightly lower level than NT-empty cells (Fig 4B). We also measured reovirus infectivity in ΔWDR81-empty and ΔWDR81-WDR81 cells using indirect immunofluorescence. The infectivity of reovirus in ΔWDR81-empty cells was significantly lower than that of NT-empty cells.

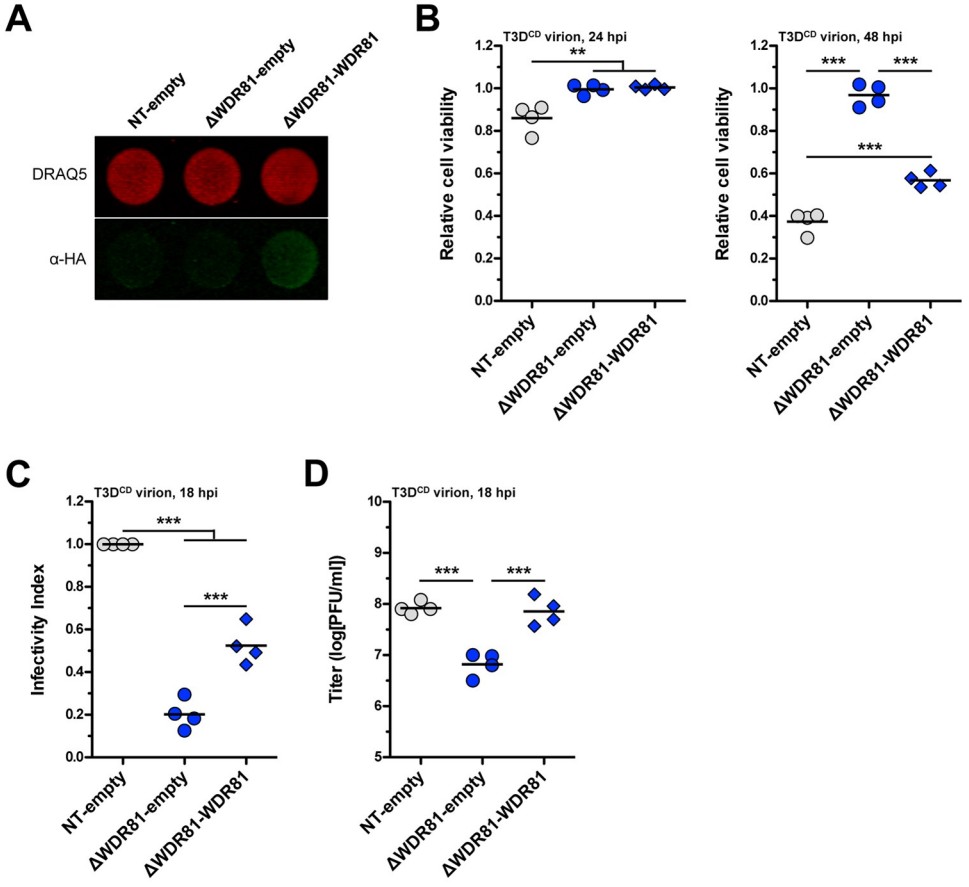

**Fig 4. Complementation of WDR81-deficient cells.** (A) WDR81 complementing cell lines. The levels of 2xHA-WDR81 in each cell line were determined by indirect immunofluorescence. The DRAQ5 (total cells) signal is false colored in red, and the IRDye 800CW (2xHA-WDR81) is false colored in green (n = 4 biological replicates; results for 1 representative experiment are shown). (B) Induction of cell death. NT-empty, ΔWDR81-empty, and ΔWDR81-WDR81 cells were infected with T3D$^{CD}$ virions at 5 PFUs/cell. At 24 and 48 h post infection (hpi), the relative cell viability was measured using the CellTiter-Glo Luminescent Cell Viability Assay. Horizontal bars indicate the means; **, $P \leq 0.005$, ***, $P \leq 0.0005$ (n = 4 biological replicates). (C) Levels of infectivity. NT-empty, ΔWDR81-empty, and ΔWDR81-WDR81 cells were infected with T3D$^{CD}$ virions at 5 PFUs/cell. At 18 hpi, the infectivity index was measured using indirect immunofluorescence. Horizontal bars indicate the means; ***, $P \leq 0.0005$ (n = 4 biological replicates). (D) Production of infectious virus. NT-empty, ΔWDR81-empty, and ΔWDR81-WDR81 cells were infected with T3D$^{CD}$ virions at 5 PFUs/cell. At 18 hpi, the amount of infectious virus produced was measured using plaque assay. Horizontal bars indicate the means; ***, $P \leq 0.0005$ (n = 4 biological replicates).

In contrast, the infectivity of reovirus was partially restored in ΔWDR81-WDR81 cells (Fig 4C). The incomplete complementation could be due to insufficient expression levels or because of less efficient interaction of hu-WDR81 with its murine partners. Nonetheless, whereas ΔWDR81-empty cells produced ~1.0 $\log_{10}$ fewer infectious particles than NT-empty cells, virus production from ΔWDR81-WDR81 cells was equivalent to NT-empty cells (Fig 4D). The phenotypic rescue of ΔWDR81 cells by the re-introduction of hu-WDR81 confirms a role for controlling an early event in reovirus replication.

## WDR81 is dispensable for infection by reovirus ISVPs

*In vitro* treatment of virions with chymotrypsin results in the degradation of σ3 and the cleavage of μ1 to δ and ϕ to form ISVPs (Fig 5A and 5B) [34]. These particles remain intact and

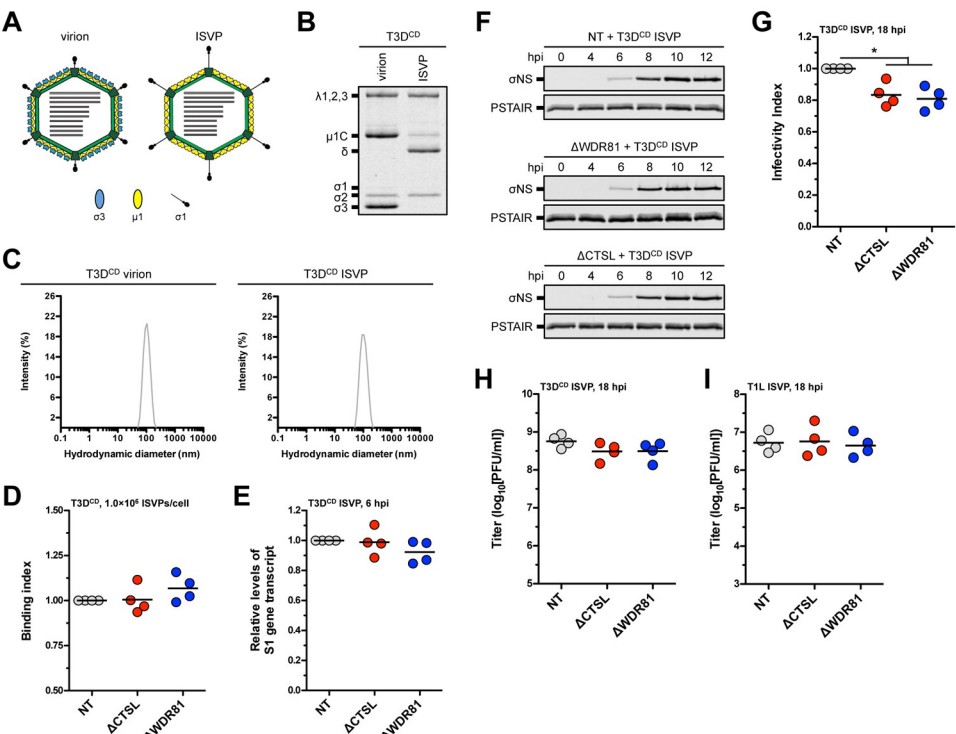

**Fig 5. Infection of WDR81-deficient cells by T3D<sup>CD</sup> or T1L ISVPs.** (A) Schematic of reovirus virions and ISVPs. (B) Protein compositions. T3D<sup>CD</sup> virions and ISVPs were analyzed by SDS-PAGE. The gel was Coomassie brilliant blue stained. The migration of capsid proteins is indicated on the left. μ1 resolves as μ1C, and μ1δ resolves as δ [74] (n = 4 biological replicates; results for 1 representative experiment are shown). (C) Size distribution profiles. T3D<sup>CD</sup> virions and ISVPs were analyzed by dynamic light scattering (n = 4 biological replicates; results from 1 representative experiment are shown). (D) Viral attachment. NT, ΔCTSL, and ΔWDR81 cells were adsorbed with T3D<sup>CD</sup> ISVPs at 1.0×10<sup>6</sup> particles/cell, and the binding index was measured using indirect immunofluorescence. Horizontal bars indicate the means (n = 4 biological replicates). (E) Viral RNA transcription. NT, ΔCTSL, and ΔWDR81 cells were infected with T3D<sup>CD</sup> ISVPs at 5 PFUs/cell. At 6 h post infection (hpi), the levels of the reovirus S1 gene transcript were measured using quantitative reverse transcription-PCR. Horizontal bars indicate the means (n = 4 biological replicates). (F) Initiation of viral protein synthesis. NT, ΔCTSL, and ΔWDR81 cells were infected with T3D<sup>CD</sup> ISVPs at 5 PFUs/cell. At the timepoints indicated in the figure, the levels of the reovirus σNS protein and the PSTAIR epitope of the host protein CDK1 were measured by Western blot. The migration of σNS and PSTAIR are indicated on the left (n = 3 biological replicates; results from 1 representative experiment are shown). (G) Levels of infectivity. NT, ΔCTSL, and ΔWDR81 cells were infected with T3D<sup>CD</sup> ISVPs at 5 PFUs/cell. At 18 hpi, the infectivity index was measured using indirect immunofluorescence. Horizontal bars indicate the means; *, $P \leq 0.05$ (n = 4 biological replicates). (H and I) Production of infectious virus. NT, ΔCTSL, and ΔWDR81 cells were infected with T3D<sup>CD</sup> or T1L ISVPs at 5 PFUs/cell. At 18 hpi, the amount of infectious virus produced was measured using plaque assay. Horizontal bars indicate the means (n = 4 biological replicates).

monodisperse in solution (Fig 5C). Moreover, *in vitro* generated ISVPs are indistinguishable from those that are formed within endosomes and, thus, are capable of launching infection [48]. Using a plate-based attachment assay, we found that ISVPs bind to the surface of NT, ΔCTSL, and ΔWDR81 cells with equivalent efficiency (Fig 5D). The S1 gene transcript accumulated to the same extent in NT, ΔCTSL, and ΔWDR81 cells (Fig 5E). Consistent with these results, the kinetics of σNS expression were similar between all cell types (Fig 5F). Compared to NT cells, the infectivity of ISVPs in ΔCTSL and in ΔWDR81 cells was only slightly lower (Fig 5G). Critically, each of the three cell types supported the production of reovirus progeny to an equivalent extent (Fig 5H). We observed comparable results when infection was initiated with ISVPs of strain T1L (Fig 5I). These data indicate that CTSL and WDR81 are dispensable

for reovirus infection that is initiated by ISVPs. The mechanisms of reovirus RNA synthesis and protein synthesis are thought to be identical, regardless of whether the infection is initiated with virions or ISVPs. Thus, our data indicate that the steps following core delivery into the host cytoplasm (Fig 2B) are not dependent on the presence of WDR81.

## Virions, but not ISVPs, accumulate in a dead-end compartment in the absence of WDR81

Because of the difference in the outcome of infection by virions and ISVPs (Figs 2 and 5), we sought to follow the fate of reovirus particles early in infection in NT, ΔCTSL, and ΔWDR81 cells. For this purpose, virions were chemically conjugated to Alexa Fluor 488 (AF488) and repurified. This labeling was most prominent on the σ3 and μ1 proteins (Fig 6A). The reovirus λ proteins were also labeled. The labeling does not affect the dispersion in solution as labeled and unlabeled particles showed an expected hydrodynamic diameter of ~100 nm (Fig 6B). Moreover, AF488-labeled particles produce plaques with the same specific infectivity as the unlabeled particles (Fig 6C). Upon infection of NT cells, AF488-labeled particles were found in cytoplasmic puncta, likely resembling endosomes. In ΔCTSL cells, even though the particles are unable to exit the endosome, the distribution of particles was similar to that observed in NT cells. In contrast, infection of ΔWDR81 cells produced puncta that appeared substantially larger and brighter, suggesting the presence of multiple particles at those sites (Fig 6D).

Similar experiments were completed with AF488-labeled ISVPs that were generated using chymotrypsin. The δ fragment of ISVP-associated μ1 retained fluorescence signal along with the λ proteins (S1A Fig). Labeled ISVPs also remained monodisperse and infectious (S1B and S1C Fig). The distribution of ISVPs within infected NT and ΔCTSL cells was similar to that observed for virions in the same cell type (Figs 6D and S1D). Interestingly, the large puncta that were observed following infection of ΔWDR81 cells with virions were no longer present (Figs 6D and S1D). The distribution of ISVPs in ΔWDR81 cells was similar to the distribution in NT and ΔCTSL cells. Together, our data support the idea that the reduced capacity of virions to launch infection is related to their accumulation in a non-productive compartment due to the absence of WDR81. Because virion-to-ISVP conversion occurs normally in WDR81-deficient cells (Fig 2D), we think that ISVPs accumulate in this dead-end compartment.

During entry, reovirus is found in early, late, and recycling endosomes [22]. Preventing particle transit through EEA1-positive early endosomes and Rab7-positive late endosomes prevents a productive infection [22]. Further, transport of reovirus to LAMP1-positive compartments prevents efficient infection [24]. To define the subcellular location where reovirus particles were trapped in the absence of WDR81, we infected NT, ΔCTSL, and ΔWDR81 cells with AF488-labeled T3D$^{CD}$ virions and determined their localization with respect to EEA1-, Rab7- and LAMP1-positive compartments. In all three cell types, we observed an equivalent proportion of incoming virions that were localized with or near EEA1 (Figs 7 and S2A). In ΔWDR81 cells, EEA1 staining also showed enlarged donut-shaped structures, which are similar to those previously observed in the absence of this protein [27]. Some proportion of virions, especially the fraction that forms large puncta, was found enclosed within this compartment. In NT and ΔCTSL cells, virions were also associated with Rab7-positive structures, which is consistent with the known transit of reovirus to late endosomes [22] (Fig 8A). Remarkably, Rab7 staining was mostly absent in ΔWDR81 cells, and therefore, few reovirus particles were found associated with Rab7 structures (Figs 8A and S2B). Reovirus particles also colocalized with LAMP1 in all cell types (Figs 8B and S2C). Similar to the EEA1 results, LAMP1-positive donut-shaped compartments were observed in ΔWDR81 cells, which also harbored virus puncta. Based on previous findings that reovirus must reach late endosomes to

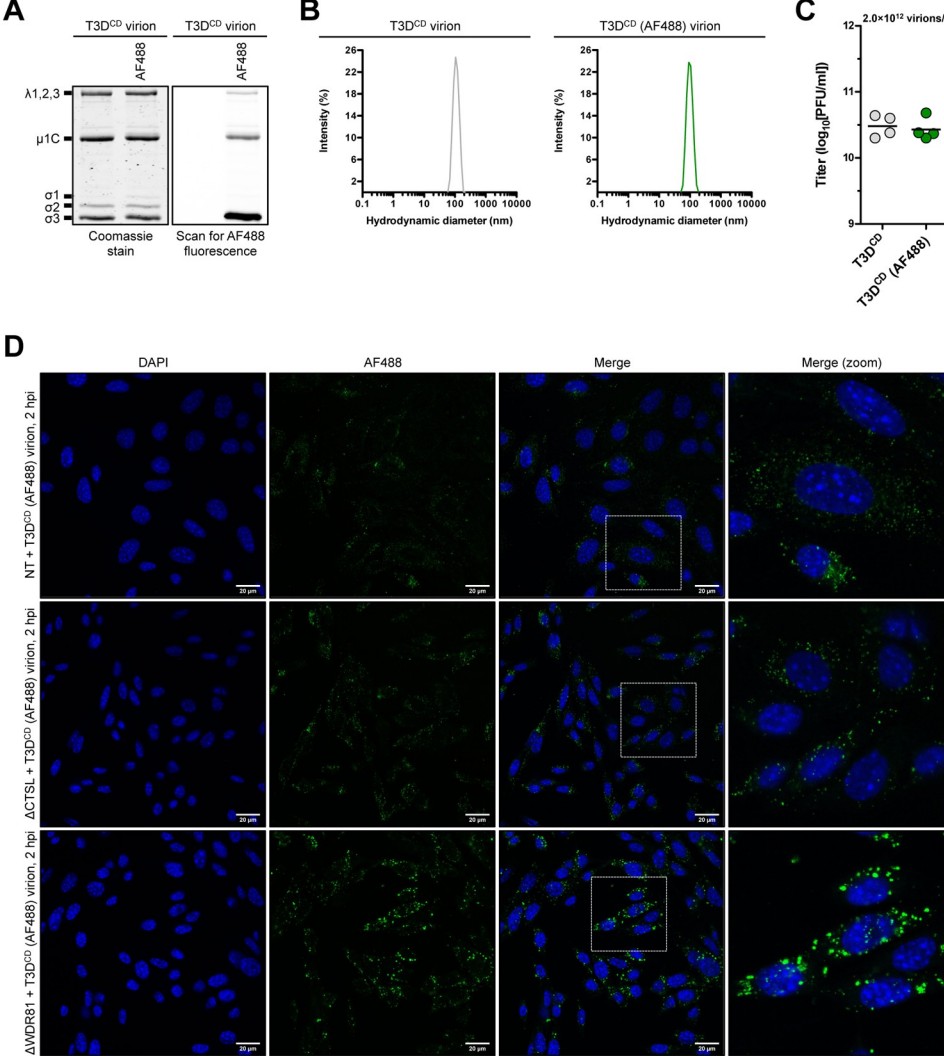

**Fig 6. Infection of WDR81-deficient cells by T3D<sup>CD</sup> (AF488) virions.** (A) Protein compositions. Unlabeled and AF488-labeled T3D<sup>CD</sup> virions were analyzed by SDS-PAGE. The gel was Coomassie brilliant blue stained (left side panel) and scanned for AF488 fluorescence (right side panel). The migration of capsid proteins is indicated on the left. μ1 resolves as μ1C [74] (n = 4 biological replicates; results for 1 representative experiment are shown). (B) Size distribution profiles. Unlabeled and AF488-labeled T3D<sup>CD</sup> virions were analyzed by dynamic light scattering (n = 4 biological replicates; results from 1 representative experiment are shown). (C) Specific infectivity. The titers of unlabeled and AF488-labeled T3D<sup>CD</sup> virions ($2\times10^{12}$ particles/ml) were determined by plaque assay. Horizontal bars indicate the means (n = 4 biological replicates). (D) Localization within an infected cell. NT, ΔCTSL, and ΔWDR81 cells were infected with T3D<sup>CD</sup> (AF488) at 10,000 virions/cell. At 2 h post infection (hpi), the infected monolayers were fixed, stained with DAPI, mounted, and imaged by confocal microscopy. The images were obtained using a X63 oil-immersion objective and processed using ImageJ software [42]. The DAPI signal is false colored in blue, and the AF488 signal is false colored in green. The inset boxes in the 'Merge' column are expanded in the 'Merge (zoom)' column. The scale bars represent 20 μm (n = 3 biological replicates; results from 1 representative experiment are shown).

launch infection [22], our data suggest that the absence of a Rab7-positive compartment in ΔWDR81 cells may account for the lack of infection. It is also possible that infection is blocked in ΔWDR81 cells because the particles are trapped in aberrant, donut like endolysosomal compartments.

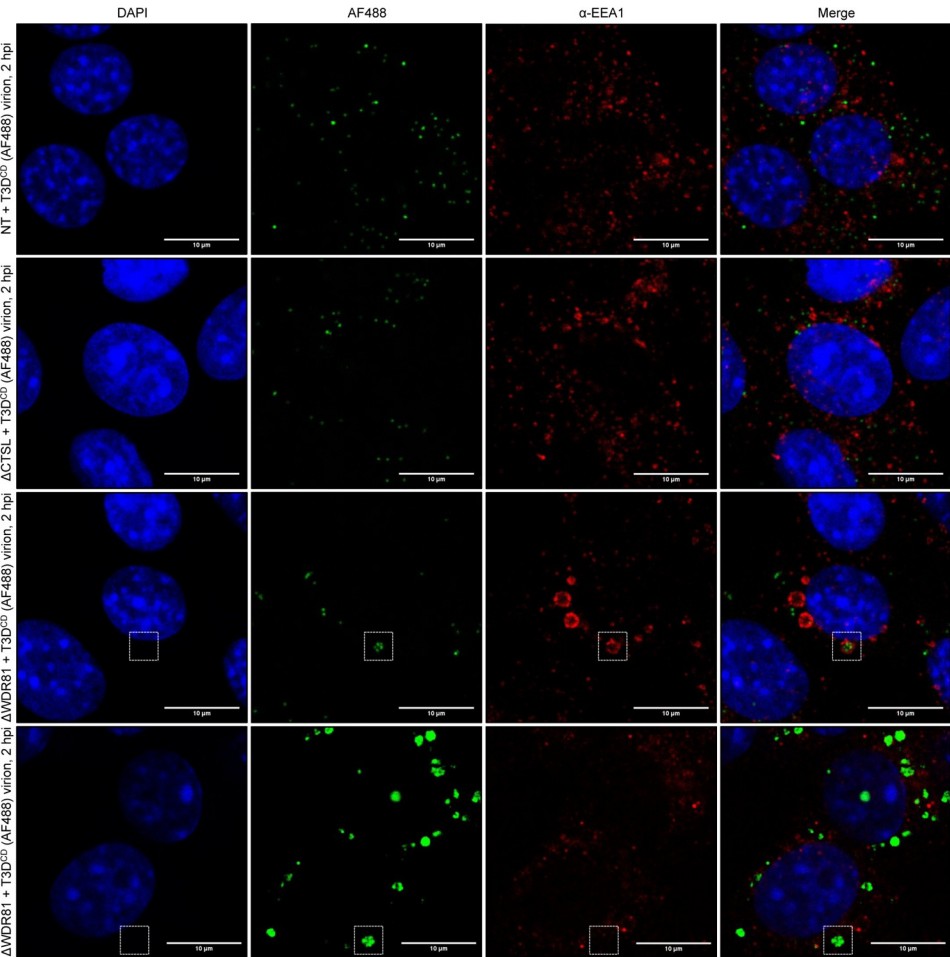

**Fig 7. Colocalization of T3D$^{CD}$ (AF488) virions with EEA1 in WDR81-deficient cells.** NT, ΔCTSL, and ΔWDR81 cells were infected with T3D$^{CD}$ (AF488) at 10,000 virions/cell. At 2 h post infection (hpi), the infected monolayers were fixed, stained with an anti-EEA1 primary antibody and with DAPI, mounted, and imaged by confocal microscopy. The images were obtained using a X63 oil-immersion objective and processed using ImageJ software [42]. The DAPI signal is false colored in blue, the AF488 signal is false colored in green, and the EEA1 signal is false colored in red. The white boxes highlight large virus puncta and/or EEA1 donut-shaped structures. The scale bars represent 10 µm (n = 3 biological replicates; results from 1 representative experiment are shown).

## WDR81 is required for infection by late penetrating viruses

Whereas reovirus virions must travel to late endosomes to launch infection, ISVPs can initiate infection either at the plasma membrane or from within early stages of the endocytic uptake pathway [18,49]. Based on our observation that infection by virions is blocked by the absence of WDR81 and infection by ISVPs is not, we hypothesized that WDR81 is required for the entry of other viruses that must transit through the late endosome to initiate infection. To test this idea, we used vesicular stomatitis virus (VSV), an early penetrating virus, and VSV that expresses the glycoprotein of Ebolavirus, a late penetrating virus, in place of the VSV G protein (VSV-EBO GP). Each of these viruses were engineered to express green fluorescent protein (GFP) in infected cells. Replacement of VSV G with EBOV GP switches the entry requirements of VSV to those needed for bona fide Ebolavirus infection [50,51].

We first tested the capacity of VSV (GFP) to compromise the viability of NT, ΔCTSL, and ΔWDR81 cells. Infection of NT cells with VSV (GFP) resulted in increasing cell death from 18

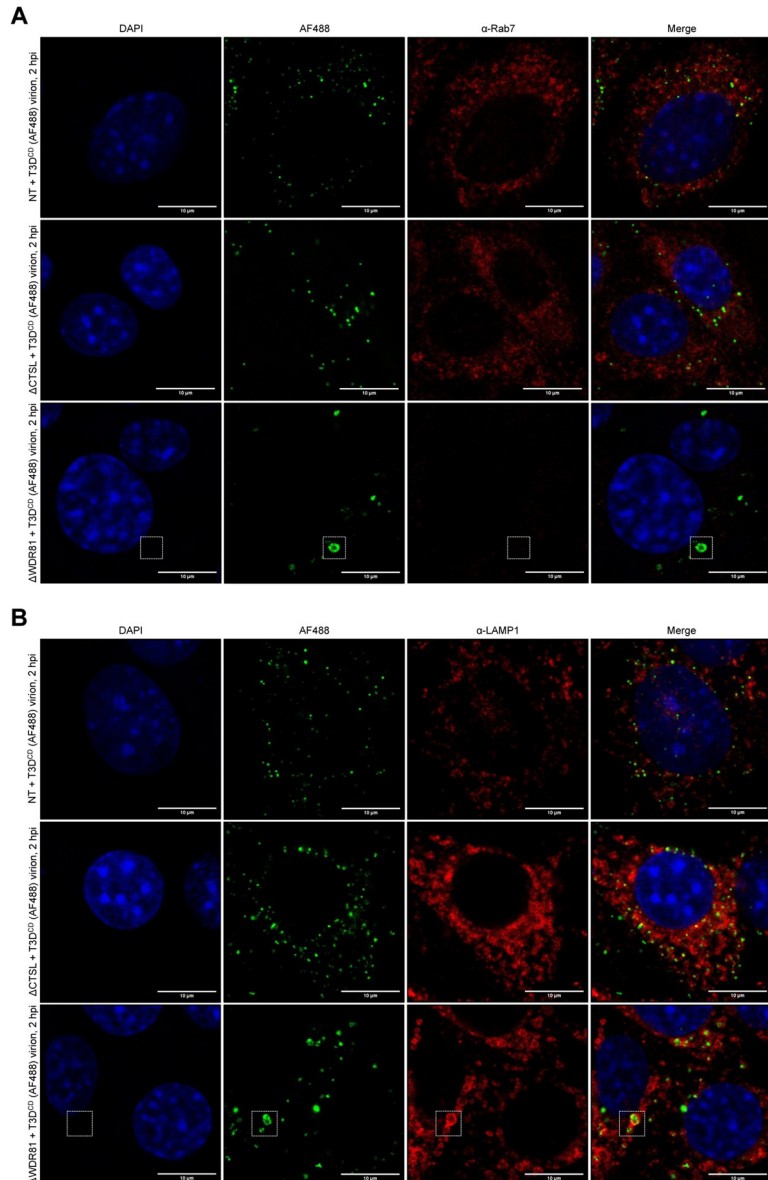

**Fig 8. Colocalization of T3D^CD (AF488) virions with Rab7 and LAMP1 in WDR81-deficient cells.** (A and B) NT, ΔCTSL, and ΔWDR81 cells were infected with T3D^CD (AF488) at 10,000 virions/cell. At 2 h post infection (hpi), the infected monolayers were fixed, stained with an anti-Rab7 (A) or anti-LAMP1 (B) primary antibody and with DAPI, mounted, and imaged by confocal microscopy. The images were obtained using a X63 oil-immersion objective and processed using ImageJ software [42]. The DAPI signal is false colored in blue, the AF488 signal is false colored in green, and the Rab7 (A) or LAMP1 (B) signal is false colored in red. The white boxes highlight large virus puncta and/ or LAMP1 donut-shaped structures. The scale bars represent 10 μm (n = 3 biological replicates; results from 1 representative experiment are shown).

to 26 hpi (Fig 9A). Whereas cell death was delayed in VSV (GFP) infected ΔCTSL and ΔWDR81 cells, the differences between NT and either CTSL- or WDR81-deficient cells were modest by 26 h. The basis for these differences in cell death kinetics was not investigated further in this study. The capacity of VSV (GFP) to establish infection in these cell types was also tested by quantifying the fraction of GFP expressing cells over time using an Incucyte S3 Live Cell Imager. Starting from equally confluent monolayers (Fig 9C), we found that VSV (GFP)

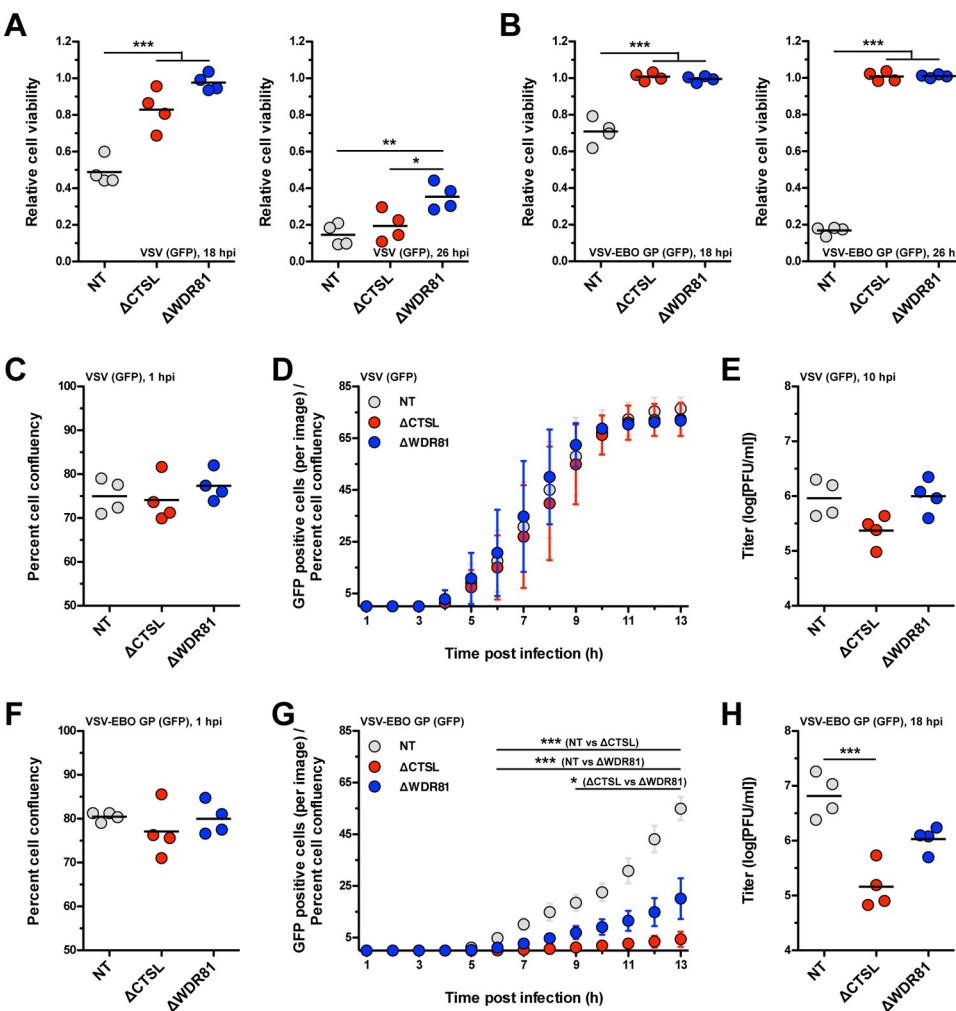

**Fig 9. Infection of WDR81-deficient cells by VSV (GFP) or VSV-EBO GP (GFP).** (A and B) Induction of cell death. NT, ΔCTSL, and ΔWDR81 cells were infected with VSV (GFP) (A) or VSV-EBO GP (GFP) (B) at 5 PFUs/cell. At 18 and 26 h post infection (hpi), the relative cell viability was measured using the CellTiter-Glo Luminescent Cell Viability Assay. Horizontal bars indicate the means; *, $P \leq 0.05$, **, $P \leq 0.005$, ***, $P \leq 0.0005$ (n = 4 biological replicates). (C, D, F, and G) Initiation of protein synthesis. NT, ΔCTSL, and ΔWDR81 cells were infected with VSV (GFP) (D) or VSV-EBO GP (GFP) (G) at 5 PFUs/cell. At the timepoints indicated in the figure, the number of total cells per image and the number of GFP positive cells per image were measured using the IncuCyte S3 Live-Cell Analysis System. The 'Percent cell confluency' at the 1 h timepoints is shown in panels C and F. Horizontal bars indicate the means (C and F); error bars indicate the standard deviations (D and G); *, $P \leq 0.05$, ***, $P \leq 0.0005$ (n = 4 biological replicates). (E and H) Production of infectious virus. NT, ΔCTSL, and ΔWDR81 cells were infected with VSV (GFP) (E) or VSV-EBO GP (GFP) (H) at 5 PFUs/cell. At 10 (E) or 18 (H) hpi, the amount of infectious virus produced was measured using plaque assay. Horizontal bars indicate the means; ***, $P \leq 0.0005$ (n = 4 biological replicates).

established infection with indistinguishable kinetics in all cell types (Fig 9D). Correspondingly, NT, ΔCTSL, and ΔWDR81 cells produced infectious progeny at similar levels (Fig 9E). Thus, WDR81 is dispensable for the replication of VSV (GFP) in MEFs.

We next tested the capacity of VSV-EBO GP (GFP) to affect the viability of NT, ΔCTSL, and ΔWDR81 cells. Whereas NT cells succumbed to VSV-EBO GP (GFP) infection within 26 h, the viability of CTSL- and WDR81-deficient cells was unaffected (Fig 9B). Cathepsin L activity is required for infection by Ebolavirus and VSV-EBO GP (GFP) [51]. Thus, the resistant phenotype of ΔCTSL cells was expected. To determine whether WDR81 is also required for infection by VSV-EBO GP (GFP), we quantified the appearance of GFP-positive cells over

time. Starting from equally confluent monolayers (Fig 9F), we observed that NT cells were efficiently infected by VSV-EBO GP (GFP). In contrast, minimal GFP-positive cells were observed in ΔCTSL cells, whereas the infectivity of VSV-EBO GP (GFP) in ΔWDR81 cells was intermediate between NT and ΔCTSL cells (Fig 9G). Consistent with this, infectious progeny production by VSV-EBO GP (GFP) was reduced in absence of either cathepsin L or WDR81 (Fig 9H). Collectively, our results suggest that viruses that travel through the late endosome share a dependence on WDR81.

## Discussion

In this manuscript, we present the results of a CRISPR-Cas9 knockout screen to identify host factors that are important for reovirus infection. We characterize the role of the endosomal protein WDR81. We find that in the absence of WDR81, reovirus particles are unable to launch viral gene expression and the viral replication program. The requirement for WDR81 is not unique to reovirus. We find that infection with VSV-EBO GP (GFP), which mimics the cell entry pathway of native Ebolavirus, is diminished by the absence of WDR81. Together, our work identifies a new host factor that is important for the entry of multiple viruses that traffic through late endosomes.

We show that infection by reovirus is dependent upon WDR81; in its absence, ISVPs are generated but do not continue the infectious cycle. These data suggest that ISVP-to-ISVP* conversion, pore formation, and/or core delivery require functional WDR81. Remarkably, these steps only require WDR81 when infection is initiated by virions. *In vitro* generated ISVPs remain capable of successfully completing virus replication [34]. Our data indicate that the requirements for the entry of *in vitro* generated ISVPs may be different. This idea is supported by another study demonstrating that endosomal cholesterol levels impact infection initiated by virions but not infection initiated by ISVPs [52]. These results counter the generally accepted assumption that once disassembled particles encounter a membrane, the remainder of the entry process (i.e., ISVP-to-ISVP* conversion, pore formation, and core delivery) is similar regardless of whether the infection was initiated by virions or ISVPs [53–56]. One possible reason for this difference may be related to the site of membrane penetration. Extracellularly produced ISVPs are no longer dependent upon disassembly by endosomal proteases and, thus, can cross the plasma membrane or the membrane of an early-stage endocytic uptake vesicle [18,49]. In contrast, virions must transit through early endosomes and reach late endosomes to successfully launch infection [22]. Because WDR81 is localized to endosomal membranes, it is possible that it directly or indirectly influences the late stages of disassembly.

Early and late endosomes are marked by their respective resident Rab GTPases. During maturation, Rab5 in early endosomes is replaced by Rab7 in late endosomes [57]. This Rab switch is accompanied by a change in the type or content of phosphoinositide species [57]. Early endosomes predominantly contain phosphatidylinositol 3-phosphate (PtdIns3P). In contrast, late endosomes contain phosphatidylinositol 3,5-bisphosphate (PtdIns(3,5)P$_2$). Each Rab GTPase, along with the phosphatidylinositol type, recruits effector proteins and, therefore, influences the timely biogenesis and function of endosomes [57]. In WDR81- or WDR91-deficient cells, the levels of PtdIns3P increase because the activity of phosphatidylinositol 3-kinase (PI3K) is no longer repressed [28]. This results in the enlargement of the early endosomal compartment. In our own experiments, we observed such large donut-shaped structures in ΔWDR81 cells that stained with EEA1 and/or LAMP1. These structures may resemble improperly generated endolysosomal compartments. Our data also suggest that in cells lacking WDR81, Rab7 containing compartments are not formed. These data agree with previous evidence suggesting that WDR81 and WDR91 are required for maturation of endosomes [28]. In

addition to WDR81 and WDR91, our screen uncovered two other proteins, Rab7 and CCZ1, that are required for endosomal maturation and reovirus infection. It is expected that the absence of Rab7, CCZ1, and WDR81 would produce similar phenotypes. As described above, the Rab5-to-Rab7 switch is a hallmark of endosome maturation. This step is stalled in the absence of Rab7 [58–60]. CCZ1 serves as a guanine exchange factor for Rab7 [61,62]. In the absence of CCZ1, Rab7 remains inactive, leading to the blockade of endosome maturation. Considering the evidence that reovirus virions must reach the late endosome for a productive infection [22], our findings suggest that perturbations prevent the formation of functional late endosomes. Based on the observation that cathepsin mediated disassembly occurs normally in ΔWDR81 cells, our data suggest that some other aspect of the late endosomal environment, such as the presence of a specific protein, a characteristic lipid environment, or a precise pH, is important for infection. Indeed, membrane lipids and pH influence ISVP-to-ISVP* conversion *in vitro* [55,56,63]. Though the role of lipid composition in ISVP* formation during infection is unexplored, when reovirus ISVPs reach an over-acidified compartment, they fail to launch infection, likely due to failure to form ISVP*s [24,63].

Our work is congruent with a parallel study, which used CRISPR-Cas9 and RNA interference screens to identify Niemann-Pick C1 (NPC1) as a putative host factor for reovirus entry [52]. NPC1 mediates the transport of cholesterol from late endosomes and lysosomes to the endoplasmic reticulum [64]. NPC1-deficient cells are unable to support the delivery of transcriptionally active reovirus core particles to the host cytoplasm, whereas treatment with hydroxypropyl-β-cyclodextrin, which binds and solubilizes cholesterol, restores infection [52]. Thus, reovirus cannot launch infection from endocytic vesicles that are disrupted by the loss of WDR81 or NPC1. Whether the phenotypes produced from each of these deficiencies are related remains to be determined.

Similar to reovirus, Ebolavirus must transit through the late endosome to initiate infection [65,66]. Following attachment to cell surface receptors, Ebolavirus traffics to the endolysosomal compartment. Within endosomes, Ebolavirus GP is cleaved by cathepsin B and L to expose the intracellular receptor binding region [51]. Cleaved GP then interacts with NPC1 within late endosomes [31,50]. This interaction is necessary for eventual fusion of the viral and host membranes. Our results indicate that VSV-EBO GP (GFP), which follows the entry pathway of native Ebolavirus [51,67], fails to efficiently initiate infection in the absence of WDR81. Because cathepsin activity in ΔWDR81 cells is sufficient to disassemble reovirus, we expect that Ebolavirus GP cleavage also occurs in these cells. As such, our hypothesis is that VSV-EBO GP (GFP) cannot launch infection because particles containing cleaved GP do not reach NPC1 positive compartments. Alternatively, infection by VSV-EBO GP (GFP) is blocked at a step following NPC1 interaction [66]. Recent CRISPR-Cas9 screens to identify host factors that are required for coronavirus infection also identified WDR81 and/or WDR91 [68–72]. The coronavirus spike protein is cleaved to become fusion competent [73]. This cleavage can occur at the plasma membrane (by the action of TMPRSS2) or within endosomes (by the action of cathepsin proteases). Work using spike mutants, which can only enter by one of these routes, suggests that WDR81-WDR91 are dispensable for plasma membrane entry but are required for endosomal entry [70]. The precise step in the coronavirus entry pathway that is dependent on WDR81-WDR91 is undefined. Nonetheless, our work demonstrates that aberrant trafficking in the absence of WDR81 is the basis for reduced infectivity for multiple late penetrating viruses.

## Supporting information

**S1 Fig. Infection of WDR81-deficient cells by T3D<sup>CD</sup> (AF488) ISVPs.** (A) Protein compositions. Unlabeled and AF488-labeled T3D<sup>CD</sup> ISVPs were analyzed by SDS-PAGE. The gel was

Coomassie brilliant blue stained (left side panel) and scanned for AF488 fluorescence (right side panel). The migration of capsid proteins is indicated on the left. μ1δ resolves as δ [74] (n = 4 biological replicates; results for 1 representative experiment are shown). (B) Size distribution profiles. Unlabeled and AF488-labeled T3D$^{CD}$ ISVPs were analyzed by dynamic light scattering (n = 4 biological replicates; results from 1 representative experiment are shown). (C) Specific infectivity. The titers of unlabeled and AF488-labeled T3D$^{CD}$ ISVPs (2×10$^{12}$ particles/ml) were determined by plaque assay. Horizontal bars indicate the means (n = 4 biological replicates). (D) Localization within an infected cell. NT, ΔCTSL, and ΔWDR81 cells were infected with T3D$^{CD}$ (AF488) at 10,000 ISVPs/cell. At 2 h post infection (hpi), the infected monolayers were fixed, stained with DAPI, mounted, and imaged by confocal microscopy. The images were obtained using a X63 oil-immersion objective and processed using ImageJ software [42]. The DAPI signal is false colored in blue, and the AF488 signal is false colored in green. The inset boxes in the 'Merge' column are expanded in the 'Merge (zoom)' column. The scale bars represent 20 μm (n = 3 biological replicates; results from 1 representative experiment are shown).
(TIF)

**S2 Fig. Quantification of colocalization between T3D$^{CD}$ (AF488) virions and EEA1, Rab7, or LAMP1 in WDR81-deficient cells.** (A-C) NT, ΔCTSL, and ΔWDR81 cells were infected with T3D$^{CD}$ (AF488) at 10,000 virions/cell. At 2 h post infection (hpi), the infected monolayers were fixed, stained with an anti-EEA1, anti-Rab7, or anti-LAMP1 primary antibody and with DAPI, mounted, and imaged by confocal microscopy. The images were obtained using a X63 oil-immersion objective and processed using ImageJ software [42]. Pearson's correlation coefficient between AF488 and EEA1 (A), Rab7 (B), or LAMP1 (C) was calculated using Just Another Colocalization Plugin (JACoP) [43]. Horizontal bars indicate the means; **, $P \leq 0.005$ (n = 3 biological replicates).
(TIF)

**S1 Data. Numerical values that were used to generate Tables and Figures in the manuscript are included. Each tab represents data for a different figure.**
(XLSX)

## Acknowledgments

We thank members of our laboratory and the Indiana University virology community for helpful suggestions. We are also grateful to Dr. Terry Dermody (University of Pittsburgh) for review of our manuscript and for sharing their unpublished results. Next generation sequencing was performed in the Indiana University Center for Genomics and Bioinformatics. Dynamic light scattering was performed in the Indiana University Physical Biochemistry Instrumentation Facility. Confocal microscopy was performed in the Indiana University Light Microscopy Imaging Center.

## Author Contributions

**Conceptualization:** Anthony J. Snyder, Pranav Danthi.

**Data curation:** Anthony J. Snyder, Andrew T. Abad.

**Formal analysis:** Anthony J. Snyder, Andrew T. Abad, Pranav Danthi.

**Funding acquisition:** Pranav Danthi.

**Investigation:** Anthony J. Snyder, Andrew T. Abad.

**Methodology:** Anthony J. Snyder.

**Project administration:** Pranav Danthi.

**Writing – original draft:** Anthony J. Snyder.

**Writing – review & editing:** Anthony J. Snyder, Pranav Danthi.

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
