## [Decision Letter · Decision Letter 0]

7 Nov 2021

Dear Dr. Danthi,

Thank you very much for submitting your manuscript "A CRISPR-Cas9 screen reveals a role for WD repeat-containing protein 81 (WDR81) in the entry of late penetrating viruses" for consideration at PLOS Pathogens. As with all papers reviewed by the journal, your manuscript was reviewed by members of the editorial board and by several independent reviewers. In light of the reviews (below this email), we would like to invite the resubmission of a significantly-revised version that takes into account the reviewers' comments.

All reviewers appreciated the attention to an interesting topic in reovrus biology but some expressed the need for more mechanistic studies. Therefore, in your revision, please pay particular attention to those studies that speak to the mechanism of action of WDR81's inhibition of reovirus entry, its location, and the identification of the cellular compartment of where it acts.

We cannot make any decision about publication until we have seen the revised manuscript and your response to the reviewers' comments. Your revised manuscript is also likely to be sent to reviewers for further evaluation.

Sincerely,

Christiane E. Wobus

Associate Editor

PLOS Pathogens

Christopher Basler

Section Editor

PLOS Pathogens

Kasturi Haldar

Editor-in-Chief

PLOS Pathogens

orcid.org/0000-0001-5065-158X

Michael Malim

Editor-in-Chief

PLOS Pathogens

orcid.org/0000-0002-7699-2064

Reviewer's Responses to Questions

**Part I - Summary**

Reviewer #1: A CRISPR-Cas9 screen reveals a role for WD repeat-containing protein 81 (WDR81) in the entry of late penetrating viruses

By AJ Snyder et al (Corresponding author: P Danthi)

Submitted to PLoS Pathog (Editorial No. PPATHOGENS-D-21-01986)

General Comments

In the search for host factors affecting different steps of orthoreovirus (reovirus) replication, a CRISPR-Cas9 knockout (KO) screen was performed on fibroblasts derived from embryonic tissue of C57/BL6 mice. Seven cellular genes were identified (WDR81, WRD91, RAB7, CCZ1, CTSL, GNPTAB, and SLC35A1) which were required for the induction of cell death following reovirus infection. The work reported here concentrated on the WD repeat-containing protein 81 (WDR81) which is involved in the maturation of endosomes (to late endosomes) required for reovirus replication. KO of WDR81 rendered cells resistant to reovirus infection. Conversely, when delta-WDR81 cells were complemented by expression of WDR81 in trans (from a transduced vector), susceptibility to reovirus infection was restored. Detailed knowledge of the different steps of the reovirus replication cycle permitted authors to localize the step of reovirus replication affected by the WDR81 KO. Since Ebolavirus requires late endosomes for its replication as well, it could be shown that the replication of a vesicular stomatitis virus (VSV)-Ebolavirus glycoprotein (GP) recombinant (labelled with GFP) was also affected by WDR81 KO (whereas replication of VSV exiting the endosomes at an early stage was not).

The Introduction on a complex subject is succinct; it could be slightly shortened. The Methods used are meticulously described. The Results are clearly presented; some of the figures shown could be improved for transparency. The Discussion does justice to the data; it could also be slightly shortened. The differentiation of reovirus infection and replication should be clarified.

This reviewer has only relatively minor additional Specific Comments.

Reviewer #2: Snyder et al conducted a screen to identify cellular genes that are required for efficient reovirus infection, using a lentivirus-derived CRISPR library transduced into murine embryonic fibroblasts. They identified seven genes whose CRISPR-mediated genetic ablation resulted in resistance to reovirus-dependent cell killing. The identification of cathepsin L, which is known to be required for the reovirus entry pathway, as one of those seven validated the use of this screening technology to identify additional required genes. The authors focused on WDR81, a protein involved in endosome maturation, and confirm that cells deficient in WDR81 are resistant to infection. Using various biochemical and genetic techniques, they show that cells lacking WDR81 fail to support reovirus infection at a step following endosomal proteolysis of the outer capsid to generate infectious subviral particles (ISVPs) and prior to nascent viral RNA synthesis, resulting in a 1.5-log reduction in infectious virus particle production. This effect was reovirus strain-independent, as both the T3DCD and T1L strains were affected by the loss of WDR81. Importantly, complementation of the WDR81-deficient cells with a lentivirus expressing wild type WDR81 restored susceptibility to reovirus infection, suggesting that the effect was specific to the genetic ablation of WDR81. Interestingly, infection with exogenously produced ISVPs was not altered by the absence of WDR81, suggesting that WDR81 facilitates cytoplasmic entry of ISVPs generated in endosomal compartments, but not ISVPs mediating host cell entry through other pathways. Confocal microscopy demonstrated accumulation of fluorophore-tagged virions, but not fluorophore-tagged ISVPs, in a subcellular “dead end” compartment in the absence of WDR81, again suggesting a defect in the endosomal entry process. Finally, the authors demonstrate that WDR81 deficiency inhibits entry of VSV pseudotyped with the EBOV glycoprotein, which requires trafficking to late endosomes for cellular entry, but has no effect on native VSV, which does not. Together, these data suggest a role for WDR81 in facilitating entry of viruses that depend on the function of late endosomes to allow for cytoplasmic delivery of viral particles.

This manuscript represents a significant step forward in the identification of host cell machinery that facilitate reovirus entry following endocytic uptake. The data are presented effectively and include appropriate controls and statistical analysis. Weaknesses of the manuscript include the lack of data to 1) identify the nature of the “dead end” compartment to which reovirus particles traffic in the absence of WDR81; and 2) determine whether reovirus particles localize to WDR81-containing endosomes during the entry process, which might at least start to give some clues about a potential mechanism. Other experiments that attempt to elucidate a mechanism for WDR81 in the reovirus entry pathway would certainly enhance the significance of the work. One final major issue has to do with the time points examined in some of the experiments. Based on the data in Fig. 2F, WDR81-deficient cells exhibit only a two hour delay in the synthesis of σNS compared to wild type cells (a strong band appearing at 12 hpi rather than 10 hpi). This is hard to reconcile with the level of decrease in infectivity and infectious titer shown at 18 hpi (as well as the much lower levels of transcription observed at 6 hpi). If reovirus protein synthesis is merely delayed by 2 h (which the authors claim is “significantly delayed”), in a single cycle replication assay (which would presumably be close to the case at an MOI of 5 PFU/cell), would the levels of virus in WDR81-deficient cells ever “catch up” to levels in wild type cells, if measured at 24 or 36 hpi? If it does not, then perhaps there are additional roles for WDR81 in facilitating cell death in response to infection (which would be more in accordance with the longer term selection of cells resistant to reovirus-induced cell death that formed the basis for the CRISPR screen). A longer time course looking at RNA levels, as well as infectivity and viral titers, would help resolve these issues.

Reviewer #3: Snyder et al., performed CRISPR-Cas9 based genome-wide screening for finding host genes related to mammalian reovirus infection. In the initial screening, several candidate genes which promote virus infections were detected along with several genes that have been studied. Among those genes that have not been investigated, authors focused on the WD repeat-containing protein 81 (WDR81) which was related to the maturation of endosomes. By using assays to evaluate virus binding and penetration, the authors concluded that without WDR81, mammalian reovirus virion is stacked inside the endosome (authors called ‘in the dead-end compartment). Lastly, the authors found that the infectivity of vesicular stomatitis virus carrying ebolavirus glycoprotein (VSV-EBO GP) but not wild-type VSV were reduced in WDR81 knock-out cells and concluded that WDR81 is important for the penetration steps of late penetrating viruses which transit through the late endosome to initiate infection. Although the manuscript is very well written and opens a new paradigm about reovirus endocytosis that remains elusive, the molecular function of WDR81 in viral infection remains to be fully elucidated. Some information should be provided, and additional experiments are required to further support the authors’ conclusions.

**Part II – Major Issues: Key Experiments Required for Acceptance**

Reviewer #1: Not applicable

Reviewer #2: 1) As outlined above, experiments to determine the nature of the “dead-end compartment” would be very useful in interpreting the nature of the block in reovirus entry. Do these compartments contain lysosomal markers? Accumulation of early endosomal markers?

2) Also as outlined above, experiments examining the location of WDR81 during reovirus entry would also aid interpretation of the potential role for WDR81 in the entry process (is it directly in the endosomal compartments containing virions, or does it exert effects through other means)?

3) A replication time course, examining both reovirus transcript levels over time as well as virus particle production over time, would aid in clarifying the potential role for WDR81 – is it merely delaying entry and protein synthesis, or does it have a role in other processes that facilitate infection?

Reviewer #3: 1. Fig 5. The author speculated that virions or processed ISVP were accumulated in the late endosome (or the dead-end compartment) in WDR81-deficient cells. Co-localization analysis of viral particles and late endosome markers (such as Rab7) is recommended to ensure that the virions were stuck inside the endosome. It is also suggested to analyze the accumulation of virions in the cytoplasm (endosome) of NT and KO cells using electron microscopy.

2. Fig 7. The author tested only two types of late penetrating viruses (reovirus and VSV-expressing Ebola G protein). Can VSV expressing ebolavirus glycoprotein be considered as a late-penetrating virus? References describing the characteristics of this virus should be included. A lot of viruses, including the dengue virus and influenza A virus, use late endosomes for cell entry. Since the title of this paper generalized that WDR81 is involved in cell entry of late penetrating virus, the test using other viruses is recommended.

**Part III – Minor Issues: Editorial and Data Presentation Modifications**

Reviewer #1: Specific Comments

Line

51 Consider reading: … establishing infection and to understand…

59f Consider phrasing: … vesicular stomatitis virus Ebolavirus glycoprotein expressing recombinant…

85 to line 103. The text could be slightly condensed.

90 … a number of host factors supporting reovirus replication have been identified…

113 … but fail to initiate further replication. [Throughout manuscript: Since part of the reovirus replication cycle has already been passed when the incoming virus particle reaches late endosomes, it appears slightly odd to speak of ‘infection’ for the further steps from ss(+)RNA transcribing core particles onwards. Rephrasing and clarification of this issue at various later places in the manuscript is suggested.]

147 … Tris-HCl, pH 7.4. [Throughout manuscript when Tris is the basis for a buffer.]

155 … AF488-labeled reovirus virions… Consider referring to the method used to produce them, lines 414ff.

156 … -tosyl-L-lysine chloromethyl ketone (TLCK)- treated chymotrypsin…

201 Omit ‘as previously described [36]’; it is repetitive (see line 197).

233 Omit ‘as previously described’.

254 … deltaWDR81-WDR81 cells… Clarify the meaning.

325 … adsorbed with 103 virions/cell… Is this correct? To how many PFUs does this number of virions correspond?

356 … delta delta CT method… Consider citation of: Livak KJ, Schmittgen TD. Analysis of relative gene expression data using real-time quantitative PCR and the 2(-Delta Delta C(T)) Method. Methods. 2001 Dec;25(4):402-8. doi: 10.1006/meth.2001.1262. PMID: 11846609.

518 … is not required for ISVP formation and that the absence of …

564 … due to insufficient expression levels or to less efficient interaction…

607 Fig. 6 should become a Supplementary figure.

626 Consider phrasing: … that expresses the glycoprotein of Ebolavirus, a late penetrating virus, in place of..

635 … not investigated further in this study…

665 … mimics the cell entry pathway of native Ebolavirus… Please provide a ref.

719 When the study of Ortega et al, 2021 (submitted) has been accepted for publication at the time of revision of this manuscript, the reference and text should be fully integrated in this manuscript.

740f Here the Discussion could be improved; the reference to coronaviruses which were not the subject of this study is not an optimal end of Discussion for this manuscript.

912ff References

Consider converting journal names to the accepted abbreviations throughout.

Lines 927-30: omit.

Figure 3, panel A. Try to improve the recognition of the (false) green fluorescence in the right well of the lower line of wells.

Figure 5, panel D. Can the brightness of the green fluorescence be improved?

Figure 6 should be considered to be transferred to Suppl. Mat. Even more than in Fig. 5, the brightness of the green fluorescence should be improved (although it is weaker in absolute terms).

Reviewer #2: 1) The CRISPR-engineered mutation causes a premature stop codon at residue 1338 of 1934, so over 2/3 of the protein is still encoded. Clearly, the protein is not functional to support reovirus infection, but what is known about the structure of the protein that might limit its activity following truncation? Are transmembrane domains missing? More information about the known domain architecture/structure of the protein would be helpful. Additionally, an immunoblot for the protein (I believe a polyclonal antiserum is commercially available?) would help to understand whether a truncated product (dominant negative?) is produced or whether the truncation leads to an unstable/abortive product.

2) Data on virus replication is presented as raw titers, not virus yield. Presenting yield instead of/in addition to virus titers may help interpret the relative impact of WDR81 on the system. Relatedly, the authors claim on line 538 that WDR81 is “required for replication” – clearly, some replication is happening, though not as efficiently as in wild type cells. (As an aside, this is, in itself, interesting – is the virus able to replicate to wild type titers in some small subset of cells that it can gain cytoplasmic access to, or is replication just less efficient or delayed in all cells, leading to the decrease in yield?).

3) Similarly, the experiments with VSV-EBO GP are not conducted to a time when replication plateaus – is replication merely delayed in the WDR81-deficient cells, or will it only infect a subset of the total (clearly the effect in the CTSL-knockouts is more significant, but it is hard to tell where the curve in the WDR81 cells would end up if allowed to progress further).

Reviewer #3: 1. Fig 2. What if the endocytosis inhibitor (such as dynasore) is used? Do the NT cells show similar infectivity to WDR81 deficient cells? What is the balance between cathepsin B and L? Does cathepsin B compensate the function of cathepsin L to induce mu1 cleavage in CTSL deficient cells? In Fig 2A, it was mentioned that the cell viability was checked at 24h and 48h post-infection in the figure legend. However, the analysis at 24h post-infection result is nowhere to be found. Please confirm the results.

2. Figs 2 and 3. In Figs 2G-H, it was revealed that the viral titer in ΔWDR81 cells was decreased in accordance with its infectivity. However, in Figs 3C-D, the reovirus infectivity of WDR81-complemented cells was still decreased, but the viral titer of WDR81-complemented cells was the same as that of NT-empty cells. I am intrigued whether the ΔWDR81 cells inhibit the reovirus infection completely or just delay the replication. Thus, I would suggest multiple step reovirus growth assay to address this. In addition, what is the reason for using human WDR81 instead of murine WDR81? The homology between those two genes is pretty high, however, infectivity in WDR81-complemented cells is still different from that in NT cells, hence it is not ideal complementation unless there is any explanation. Please provide any reasoning regarding this decision.

3. Fig 5. Why is the virus not accumulated in the late endosome in CTSL deficient cells? As mentioned in the manuscript, the virions are not able to be converted to ISVP in the CTSL deficient cells, resulting in the failure of viral penetration. The author should give a discussion on that part.

4. Line 85. .. is a segmented double-stranded RNA (dsRNA) virus..

5. The author identified several genes including WDR81, WDR91, and CTSL that were required for the reovirus infection by CRISPR-Cas9 screening and focused on the role of WDR81 (and CTSL) in reovirus infection. The author also mentioned that WDR81 and its binding partner WDR91 play an important role in endosomal maturation. Thus, the author might want to include the data indicating the importance between WDR81 and WDR91 interaction in reovirus infection if possible.

PLOS authors have the option to publish the peer review history of their article (what does this mean?). If published, this will include your full peer review and any attached files.

Reviewer #1: No

Reviewer #2: No

Reviewer #3: No
---

## [Editor Report · Decision Letter 1]

17 Feb 2022

Dear Dr. Danthi,

Thank you very much for submitting the revised version of your manuscript "A CRISPR-Cas9 screen reveals a role for WD repeat-containing protein 81 (WDR81) in the entry of late penetrating viruses" for consideration at PLOS Pathogens. Your response to reviewers comments and revised manuscript was reviewed by members of the editorial board. We appreciate the detailed and thoughtful response and inclusion of new experimental data in Figures 7 and 8. However, Fig. 7 and 8 provide representative images of immunofluorescence images but lacks quantification across different experiments to support statements in the text. For example: What is the proportion of virions colocalized with EEA1 in the various conditions in Fig. 7? In Figure 8, what proportion of virus was colocalized with Rab7 and Lamp1 in the different conditions? 

We are likely to accept this manuscript for publication provided that you modify the manuscript accordingly by adding quantification to Figures 7 and 8 and making any necessary edits to the manuscript.

Sincerely,

Christiane E. Wobus

Associate Editor

PLOS Pathogens

Christopher Basler

Section Editor

PLOS Pathogens

Kasturi Haldar

Editor-in-Chief

PLOS Pathogens

orcid.org/0000-0001-5065-158X

Michael Malim

Editor-in-Chief

PLOS Pathogens

orcid.org/0000-0002-7699-2064

Reviewer Comments (if any, and for reference):

Figure Files:

Data Requirements:

Reproducibility:

References:

---

## [Editor Report · Decision Letter 2]

25 Feb 2022

Dear Dr. Danthi,

We are pleased to inform you that your manuscript 'A CRISPR-Cas9 screen reveals a role for WD repeat-containing protein 81 (WDR81) in the entry of late penetrating viruses' has been provisionally accepted for publication in PLOS Pathogens.

Best regards,

Christiane E. Wobus

Associate Editor

PLOS Pathogens

Christopher Basler

Section Editor

PLOS Pathogens

Kasturi Haldar

Editor-in-Chief

PLOS Pathogens

orcid.org/0000-0001-5065-158X

Michael Malim

Editor-in-Chief

PLOS Pathogens

orcid.org/0000-0002-7699-2064
---

## [Editor Report · Acceptance letter]

9 Mar 2022

Dear Dr. Danthi,

We are delighted to inform you that your manuscript, "A CRISPR-Cas9 screen reveals a role for WD repeat-containing protein 81 (WDR81) in the entry of late penetrating viruses," has been formally accepted for publication in PLOS Pathogens.

Best regards,

Kasturi Haldar

Editor-in-Chief

PLOS Pathogens

orcid.org/0000-0001-5065-158X

Michael Malim

Editor-in-Chief

PLOS Pathogens

orcid.org/0000-0002-7699-2064